# CMC-Bench: Towards a New Paradigm of Visual Signal Compression

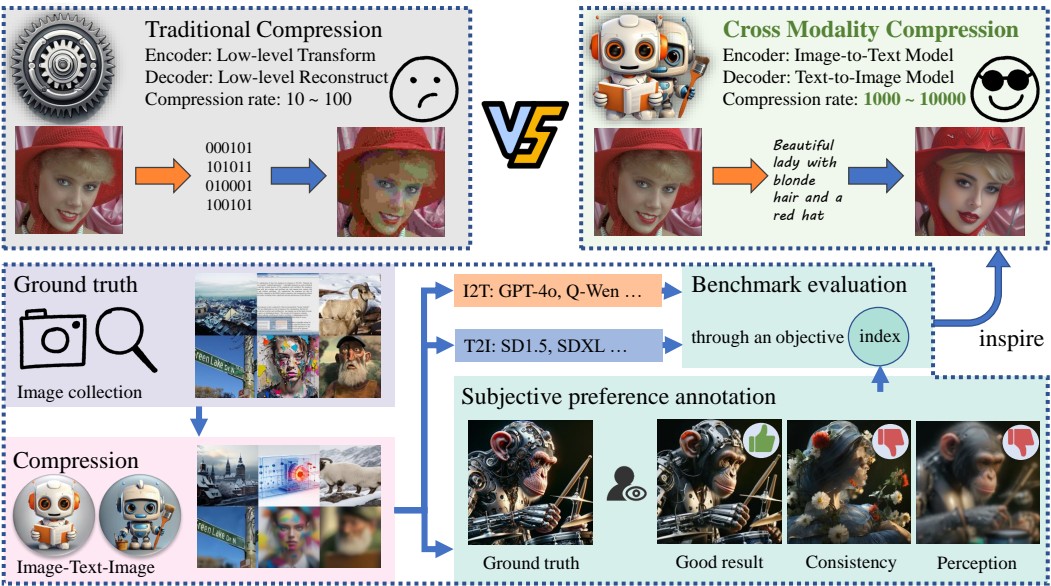

Figure 1: Overview of CMC-Bench. We demonstrate the superiority of Cross Modality Compression over traditional codecs, and subjective and objective evaluations of compression results on Consistency and Perception. This benchmark can motivate it to become the future codec paradigm.

## Abstract

Ultra-low bitrate image compression is a challenging and demanding topic. With the development of Large Multimodal Models (LMMs), a Cross Modality Compression (CMC) paradigm of Image-Text-Image has emerged. Compared with traditional codecs, this semantic-level compression can reduce image data size to 0.1% or even lower, which has strong potential applications. However, CMC has certain defects in consistency with the original image and perceptual quality. To inspire insights into such a problem, we introduce CMC-Bench, a benchmark of the cooperative performance of Image-to-Text (I2T) and Text-to-Image (T2I) models for image compression. This benchmark covers 18,000 and 40,000 images respectively to verify 6 mainstream I2T and 12 T2I models, including 160,000 subjective preference scores annotated by human experts. At ultra-low bitrates, it proves that the combination of some I2T and T2I models has surpassed the most advanced visual signal codecs; meanwhile, it highlights where LMMs can be further optimized toward the compression task. We encourage LMM developers to participate in this test to promote the evolution of visual signal codec protocols.

## 1 Introduction

Visual signal compression aims to minimize image data, playing a crucial role in delivering high-quality image/video services with limited network resources and storage capacity. Since the MPEG-1 (Le Gall, 1991) standard was introduced, compression rates for visual signals have doubled (Skodras et al., 2001; Wiegand et al., 2003; Sullivan et al., 2012; Bross et al., 2021) every decade. In

recent years, traditional image codecs have achieved a 500 times compression rate while ensuring a decent visual experience for humans. However, traditional codecs are approaching the Shannon limit of 1,000 times Compression Rate (CR) in the upcoming next-generation protocols. Fortunately, the rapid development of Large Multimodal Models (LMMs) has opened up possibilities for such Ultra-low-Bitrate (ULB) compression.

**Why use LMMs for compression?** LMMs support the conversion between multiple modalities, where text consumes much less space than image modalities. By cascading Image-to-Text (I2T) and Text-to-Image (T2I) models, images can be compressed and reconstructed from semantic information. This Cross-Modality Compression (CMC) paradigm operates at the semantic level, which outperforms traditional codecs at the pixel level. It enables easy attainment of ULB, and even the Extreme-low Bitrate (ELB) for CR about 10,000 times. However, at such low bitrates, CMC presents two significant issues that cannot be overlooked. (1) **Consistency**: The reconstruction process heavily relies on intermediate textual information. Any omission by the I2T model (encoder) or misunderstanding by the T2I model (decoder) can result in severe distortion. Unlike minor changes in brightness and color, this can lead to a semantic-level inversion of the entire image. (2) **Perception**: Textual encoding provides a coarse representation of the image, necessitating the T2I model to add details. Insufficient details degrade perceptual quality, while excessive ones compromise faithfulness to the original image. Unfortunately, as the bitrate decreases, the conflict between them (Blau & Michaeli, 2018; 2019) becomes more pronounced. As the consistency and perception failure cases in Figure 1, these issues jointly limit the application of CMC.

For LMMs, there is a lack of effective evaluation criteria both in terms of consistency and perceptual aspects. Although numerous benchmarks have recently emerged for LMMs, they are primarily designed to assess the performance of either I2T (Image-to-Text) or T2I (Text-to-Image) models working alone, such as captioning/visual question answering for I2T (Liu et al., 2024b; Li et al., 2023a), or generation quality/realism for T2I (Bakr et al., 2023; Huang et al., 2023). Consequently, we introduce the first joint benchmark called CMC-Bench, aimed at testing the collaborative capabilities of I2T and T2I models. Our contributions include:

- A large-scale dataset consists of 58,000 images using the CMC paradigm. 4,000 images among them have 160,000 expert annotations, covering both consistency and perception issues, paving the way for information loss modeling in the I2T and T2I processes.

- A comprehensive evaluation standards, consisting of four compression modes under different requirements, along with the two dimensions mentioned above. We validate mainstream models (including 6 I2T and 12 T2I) to explore optimal combinations.

- A throughout comparison with traditional codecs. We compared the benchmark winner with existing image codecs, revealing the significant advantages of the CMC image compression paradigm and some remaining drawbacks. We encourage LMM developers (both I2T and T2I) to participate in CMC-Bench to further expand the application of CMC.

## 2 RELATED WORKS

**Cross-Modality Compression.** The earliest CMC method (Li et al., 2021) emerged in 2021, achieving a compression ratio of almost 10,000 times through text modality. However, as a simple combination of I2T and T2I models, their results often exhibit noticeable differences from the original images. Subsequently, Text+Sketch (Lei et al., 2023) employed edge operators and ControlNet (Zhang et al., 2023a) to refine CMC, but its consistency remained inferior to traditional codecs. The most advanced CMC methods, like M-CMC and MISC (Gao et al., 2024b;a; Mao et al., 2024; Xue et al., 2024; Li et al., 2024a;c), have surpassed advanced codecs like VVC (Bross et al., 2021) in both consistency and perception, indicating the promising future of this paradigm. Nevertheless, there is still room for improvement in these two aspects. All existing CMC methods are from one specific I2T and one T2I model, and the models used are relatively outdated. Considering the rapid development of Generative-AI, how to combine the latest models towards a better CMC becomes an unrevealed question.

**Benchmark for LMM Evaluation.** Existing benchmarks are mainly designed for T2I and I2T models. For I2T, they usually take a specific image sequence as input, compare the text output by LMM with the ground truth, and use the relevance of the two as a performance indicator. The

Table 1: Image compression datasets with subjective label. Keys [Ref: Reference, Dis: Distorted]

| Dataset | Ref | Dis | Ratings | Score | Resolution | Image type | Dimension |
|---|---|---|---|---|---|---|---|
| CLIC2021 | 585 | 2,730 | 122,107 | DS | 768 | NSI | Consistency |
| CLIC2022 | 585 | 2,730 | 57,300 | DS | 768 | NSI | Consistency |
| NTIRE2022 | 250 | 29,150 | 1,880,000 | MOS | 288 | NSI | Consistency, Perception |
| SCID | 40 | 1,800 | 18,000 | MOS | 1,280 | SCI | Consistency |
| CCT | 72 | 1,320 | 26,400 | MOS | 1,280 | NSI, SCI | Consistency |
| AGIQA-3K | - | 2,982 | 62,622 | MOS | 512 | AIGI | Perception |
| ImageReward | - | 136,892 | 136,892 | SS | 512~1,024 | AIGI | Perception |
| **CMC-Bench** | **1,000** | **58,000** | **160,000** | **MOS** | **512~1,024** | **NSI, SCI, AIGI** | **Consistency, Perception** |

annotation content includes common sense (Liu et al., 2024b; Li et al., 2023a) or specific expert fields (Wu et al., 2024; Li et al., 2024h; Zhang et al., 2024a; Wu et al., 2023a). For T2I, the input is a carefully designed text prompt (Bakr et al., 2023; Huang et al., 2023) (e.g. different themes, adjectives, and spatial relationship). They use specific visual encoders to process the output image of LMM and determine its alignment with the text as the generative performance (Saharia et al., 2022; Cho et al., 2023). However, as the current CMC paradigm is still immature, there is no pipeline for the joint evaluation of I2T+T2I model.

**Benchmark for Image Compression.** Given the significance of visual information compression, several related competitions (Ballé et al., 2020; Gu et al., 2022) have been held in recent years. However, these competitions often limit their scope to Natural Scene Images (NSIs). Screen Content Images (SCIs) (Ni et al., 2017; Min et al., 2017), which are prevalent on the internet, and the emerging AI-Generated Images (AIGIs) (Li et al., 2023c; Xu et al., 2024; Zhang et al., 2023c) have received some attention with new datasets, but no existing dataset comprehensively considers them together. Moreover, the performance evaluation of compression algorithms can be challenging, often requiring subjective quality assessments from human viewers to train Image Quality Assessment (IQA) (Li et al., 2023b; 2022; Zhang et al., 2024c;d; 2023f;d; Li et al., 2024f; Zhang et al., 2024b) models, which provide objective metrics for compression algorithms. In the context of ULB image compression, both the **consistency** between the distorted and reference images, as well as the inherent appeal of the distorted image in human **perception**, need to be annotated. Existing IQA datasets typically annotate only one aspect, while often in a coarse-grained manner through Single Stimulus (SS) or Double Stimulus (DS) comparisons. In contrast, Mean Opinion Score (MOS) derived from multiple subjects offers a more detailed and objective evaluation as shown in Table 1.

## 3 DATASET CONSTRUCTION

### 3.1 GROUND TRUTH SELECTION

To provide a comprehensive and high-quality resource for various applications on the Internet, we carefully curated 1,000 high-quality images without compression distortion as the ground truth of CMC-Bench. Among them, NSIs are the most mainstream content, so we selected 400 images. At the same time, considering that SCIs are more common on screens and AIGIs are increasing on the Internet in the upcoming LMM era, we selected 300 images from each of these two categories. The specific content is as shown in Figure 2.

**NSI**: A collection of 200 high-quality Professional Generated Content (PGC) released by TV stations and photographers, specifically sampled from the CLIC database (Ballé et al., 2020); and 200 User Generated Content (UGC) by average users, selected from MS-COCO (Lin et al., 2014). To ensure image quality, we employed Q-Align (Wu et al., 2023b) to filter out low-quality UGC that might be overexposed.

**SCI**: Consisting 100 computer graphics from CGIQA-6K (Zhang et al., 2023e) in animated movies; 100 game renders from CCT and CGIQA-6K (Min et al., 2017; Zhang et al., 2023e); and 100 webpages with both images and text from CCT, SCID, and Webpage Saliency datasets (Min et al., 2017; Ni et al., 2017; Shen & Zhao, 2014). To maintain frame clarity, we also applied Q-Align (Wu et al., 2023b) to address factors like motion blur that affect visual quality.

**AIGI**: Comprises 50 images each, generated by 6 latest models: DALLE3, MidJourney, PG v25, PixArt $\alpha$, SDXL, and SSD-1B (Ramesh et al., 2022; Holz, 2023; Li et al., 2024g; Chen et al.,

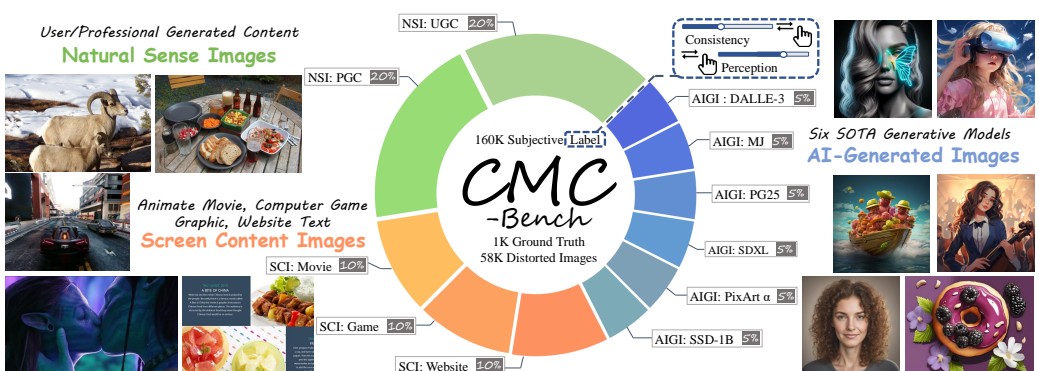

Figure 2: Source data illustration of CMC-Bench from three content types.

2023a; Rombach et al., 2022b; Gupta et al., 2024). They have demonstrated exceptional preference in previous subjective ratings (Li et al., 2024b; Liu et al., 2024a; Kou et al., 2024; Li et al., 2024d;e), representing the pinnacle of AIGI capabilities.

## 3.2 COMPRESSION MODE

Drawing on previous work in CMC, we categorize CMC into four working modes, as shown in Figure 3. Each type employs distinct configurations and is suitable for different scenarios:

**Text**: The I2T model converts images to text and is directly restored by the T2I model. Due to its reliance on the text modality only, this approach achieves a CR of 10,000, ideal for ELB situations.

**Pixel**: Each $64 \times 64$ blocks from ground truth are merged and quantized into one pixel. Beyond the *Text* mode, these pixels initialize the T2I process. The pixel representation is relatively compact, offering a CR of around 5,000, suitable for less rigorous ELB but higher demands on consistency.

**Image**: Traditional codecs are employed to compress the image, which serves as input for the T2I model for enhancement. Unlike the previous two, it omits the time-consuming I2T process by leaving the text input of the T2I model empty. This approach can achieve a CR of 1,000, suitable for ULB bandwidth but with high real-time requirements.

**Full**: Extending the *Image* mode, the T2I is guided by text content, encompassing the full pipeline of I2T, traditional codec, and T2I. It also has a CR of approximately 1,000, suitable for the most demanding performance scenarios.

## 3.3 BENCHMARK CANDIDATES

We employ 6 I2T and 12 T2I models across four compression modes. Due to the absence of text, the I2T model is not used in the *Image* mode; while for T2I, among the 4 Image Reconstruction (IR) models requiring an initial image and are not compatible with *Text* and *Pixel* modes. The remaining 8 T2I generative models support all modes. We use one certain T2I, and validate all possible I2T models to verify their performance separately (vice versa for T2I validation). For a fair comparison, We fixed RealVis (Civital, 2024) to minimize the T2I process distortion, which ensures the performance fluctuation mainly comes from the I2T. Similarly, we fix I2T as GPT-4o (OpenAI, 2023) when validating T2I models. Each I2T model produces 3,000 images, while restorative and generative models for T2I have 2,000 and 4,000, respectively. A total of $18,000 + 40,000 = 58,000$ images are generated.

I2T model: GPT-4o (OpenAI, 2023), LLAVA-1.5 (Liu et al., 2023), MPlugOwl-2 (Ye et al., 2023), Qwen (Bai et al., 2023), ShareGPT (Chen et al., 2023b), and InstructBLIP (Dai et al., 2023). Except for one model (Dai et al., 2023) for image captioning with default token length, we modify the output length of others to 10~20 words for a balance between bitrate and performance.

T2I model: Animate (Guo et al., 2024), Dreamlike (dreamlike art, 2023), PG20 (PlaygroundAI, 2023), PG25 (Li et al., 2024g), RealVis (Civital, 2024), SD15 (Rombach et al., 2022a), SDXL (Rombach et al., 2022b), and SSD-1B (Gupta et al., 2024) as generative model; DiffBIR (Lin et al.,

Figure 3: Illustration of 4 working modes of CMC. *Text* mode roughly reconstructs the semantic information, *Pixel* mode slightly improves low-level consistency, *Image* mode provides a similar structure towards ground truth but a different character, and *Full* mode has the best performance.

2024), InstructPix (Brooks et al., 2023), PASD (Yang et al., 2024), and StableSR (Wang et al., 2023b) as IR model. A higher denoising strength indicates a more obvious modification on the starting point. To balance the consistency and perception indicators, we set the strength of *Full* and *Image* modes as 0.5, the *Pixel* mode as 0.8, and the *Text* mode as the default 1.

Traditional codec: For *Full* and *Image* mode, we utilize the most advanced traditional codec VVC (Bross et al., 2021) to provide a reference image. Towards 1,000 times compression, we take its nearest bitrate that meets the ULB requirement, where the Quantizer Parameter (QP) is 53.

### 3.4 HUMAN PREFERENCE ANNOTATION

Referring from previous large-scale subjective annotation (Ballé et al., 2020) methods, we do not perform coarse-grained labeling on the entire dataset considering the limitation on annotator numbers. Instead, we fine-grain the annotations on 4,000 images to ensure multiple ratings for each image. Note that, as the benchmark indicator should be adjusted on subjective data, we did not directly select subsets from the 58,000 test images. Instead, we generated new images to prevent prior exposure to the content being evaluated. Given the greater impact of T2I models on CMC tasks than I2T models, we follow the T2I paradigm described in Section 3.3. The I2T model is fixed as GPT-4o (OpenAI, 2023) and combined with 12 different T2I models, compressing 100 ground truth into 4,000 distorted images. To ensure quality diversity, we randomly assigned strength from 0.2 to 0.9 rather than a fixed value. Each distorted image is paired with its corresponding ground truth and shown to 20 experienced participants who provided ratings on consistency and perception dimensions. Each image is then summarized into two overall scores from 0 to 5, combining all participants' feedback. For a detailed description of the experimental setup and data process, please refer to the appendix.

## 4 EXPERIMENT

### 4.1 EVALUATION INDICATOR SETTINGS

All 6 I2T and 12 T2I LMMs are verified and tested by different parameters and fixed them towards an optimal situation according to Section 3.3, while the internal model weight remains zero-shot to ensure fairness in ranking.

***Image-To-Text Configuration***: Towards different output lengths, we applied different prompts as the input for I2T models. The prompt follows previous CMC templates (Li et al., 2024c), with length [5, 10, 20, 50]. For example, to describe Figure 4 in different lengths, the input prompt format and output text from GPT-4o (OpenAI, 2023) are:

Q(5): *'Generate an informative paragraph in 5 words based on the image.'*
A(5): *'Traditional attire on serene river.'*

Q(10): *'Generate an informative paragraph in 10 words based on the image. Considering Light, Color.'*
A(10): *'Two individuals navigate a wooden boat in traditional attire, serene.'*

Q(20): *'Generate an informative paragraph in 20 words based on the image. Considering Light, Color, Clarity.'*
A(20): *'A woman in traditional attire and a conical hat, seated on a wooden boat, with another person rowing on water.'*

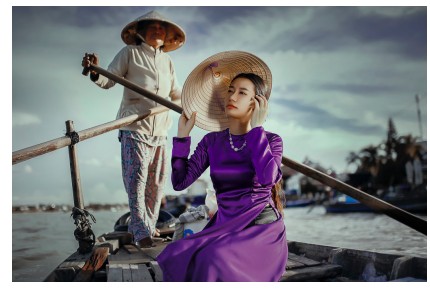

Figure 4: Example ground truth.

Q(50): *'Generate an informative paragraph in 50 words based on the image. Considering Light, Color, Clarity, Dense Caption, and Region Semantic.'*
A(50): *'A woman in a purple dress sits near the front of a boat, her hand adjusting a straw hat. Behind her, a person in a beige top and floral pants stands near the boat's wooden oars. The setting is a calm, expansive body of water under a cloudy sky.'*

According to the appendix verification, for each model, the output length is set closest to this bitrate, namely Q(20) for GPT-4o (OpenAI, 2023), MPlugOwl-2 (Ye et al., 2023); Q(10) for LLAVA (Liu et al., 2023), Qwen (Bai et al., 2023), ShareGPT (Chen et al., 2023b); and the default length for InstructBLIP (Dai et al., 2023). The temperature is zero and over-lengthed output will be cut.

***Text-To-Image Configuration***: In *Text* mode, since there is no reference image as a starting point, the denoising strength is undisputedly 1. In the other three modes, we adjusted different intensities with a granularity of 0.1. For *Full* and *Image* that provide a reference image, a high denoising strength will waste the reference information, so we verified the performance under strength from 0.2 to 0.8; for *Pixel* mode, since the pixel provides less information than the compressed image, we increased the strength and range from 0.4 to 0.99 (as strength=1 will ignore the reference). According to the appendix verification, we set the strength of *Full* and *Image* mode to 0.5 and the *Pixel* mode to 0.8.

Taking the reference and distorted image pairs as input, We use TOPIQ (Chen et al., 2024), the most advanced IQA metric in Full-Reference (FR) and No-Reference (NR) configuration to characterize consistency and perception. The average score of 1,000 ground truth images is reported as the final performance. Combining these two issues towards 4 working modes, the models are evaluated by 8 indicators for generative T2I, 6 indicators (exclude *Image* mode) for I2T, and 4 indicators (exclude *Pixel* and *Text* mode) for T2I restorative models. A weighted average of $2\times$ FR indicators and $1\times$ NR indicators is given as the overall score for ranking since the TOPIQ-FR has a smaller floating range than TOPIQ-NR. Such weight ratio can reach a balance between consistency and perception. As restorative models only support ULB compression in *Full* and *Image* mode, but not ELB compression in *Pixel* and *Text* mode, the overall score of the T2I model is ranked under ULB and ELB respectively.

In addition to TOPIQ, we also used four cutting-edge FR-IQA (AHIQ (Lao et al., 2022), DISTS (Ding et al., 2020), LPIPS (Zhang et al., 2018a), PieAPP (Prashnani et al., 2018)) and NR-IQA (CLIPIQA (Wang et al., 2023a), CN-NIQA (Kang et al., 2014), DBCNN (Zhang et al., 2018b), HyperIQA (Su et al., 2020)) algorithms to objectively score the distorted images in terms of consistency and perception. The higher the Spearman ($\sigma$) and Kendall

Table 2: Correlation between objective IQA evaluation and subjective human preference.

| Consistency | $\sigma \uparrow$ | $\kappa \uparrow$ | Perception | $\sigma \uparrow$ | $\kappa \uparrow$ |
|---|---|---|---|---|---|
| AHIQ | 0.844 | 0.645 | CLIPIQA | 0.825 | 0.623 |
| DISTS | 0.795 | 0.599 | CNNIQA | 0.584 | 0.414 |
| LPIPS | 0.583 | 0.406 | DBCNN | 0.833 | 0.640 |
| PieAPP | 0.433 | 0.294 | HyperIQA | 0.730 | 0.534 |
| **TOPIQ** | **0.943** | **0.792** | **TOPIQ** | **0.901** | **0.738** |

Rank-order Correlation Coefficient ($\kappa$), the better correlation between the objective and subjective scores. All models are trained on 80% of the distorted images in Section 3.4 and tested on the remaining 20%. Experiments in Table 2 show that the correlation between the fine-tuned TOPIQ (Chen et al., 2024) and the subjective score is outstanding with $\sigma$ beyond 0.9 in both dimensions, making it appropriate performance indicators reflecting human preference for compressed images.

The training of FR/NR quality indicators is conducted on an online server with 4 NVIDIA A6000 GPUs. The inference of I2T encoding and T2I decoding is based on a local NVIDIA GeForce

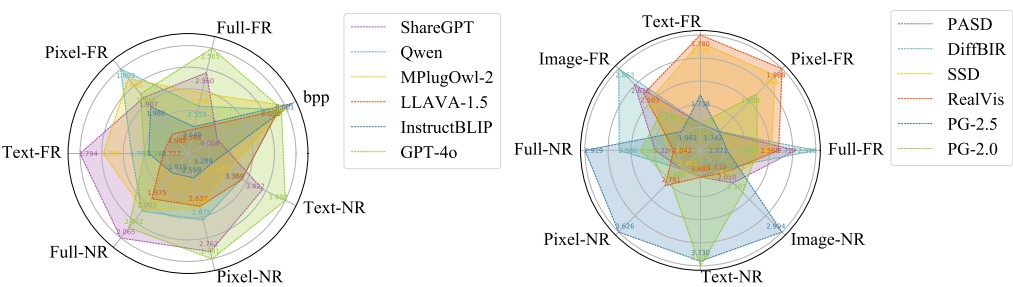

Figure 5: A radar map illustrates the collaboration of mainstream I2T (left) and T2I (right) LMMs. The model are tested as {6 different I2Ts + RealVis (Civital, 2024)} and {GPT-4o (OpenAI, 2023) + 12 different T2Is}. Only 6 T2Is with the best performance are shown in the radar map.

Table 3: Benchmark result in *Full* and *Image* modes for 8 T2I generative and 4 restorative models at ultra-low bitrate (1,000 times compression). NSI/SCI/AIGI stands for the compressed image types. FR and NR indicate consistency and perception scores. [Key: **Best**; Second Best]

| Index | Full-FR ↑ | | | Full-NR ↑ | | | Image-FR ↑ | | | Image-NR ↑ | | | Overall ↑ |
|---|---|---|---|---|---|---|---|---|---|---|---|---|---|
| T2I Model | NSI | SCI | AIGI | NSI | SCI | AIGI | NSI | SCI | AIGI | NSI | SCI | AIGI | |
| DiffBIR | **3.052** | **2.785** | **2.877** | **2.778** | **2.380** | 2.517 | **2.899** | **2.804** | **2.873** | 1.847 | 1.644 | 1.674 | **2.647** |
| PASD | 2.796 | 2.621 | 2.741 | 2.339 | 1.932 | 2.367 | 2.652 | 2.583 | 2.675 | 2.056 | 1.818 | 2.141 | 2.494 |
| RealVis | 2.617 | 2.475 | 2.584 | 1.914 | 1.808 | 2.445 | 2.509 | 2.441 | 2.558 | 1.686 | 1.675 | 2.110 | 2.331 |
| PG25 | 2.145 | 1.922 | 2.123 | 2.730 | 2.582 | 3.509 | 1.952 | 1.895 | 2.040 | 2.750 | 2.852 | 3.459 | 2.330 |
| SSD-1B | 2.515 | 2.407 | 2.554 | 1.905 | 1.878 | 2.516 | 2.386 | 2.355 | 2.512 | 1.758 | 1.783 | 2.309 | 2.305 |
| PG20 | 2.435 | 2.245 | 2.376 | 2.200 | 2.072 | 2.893 | 2.263 | 2.174 | 2.301 | 2.011 | 1.992 | 2.683 | 2.299 |
| StableSR | 2.599 | 2.591 | 2.688 | 1.401 | 1.373 | 1.549 | 2.576 | 2.582 | 2.679 | 1.392 | 1.367 | 1.541 | 2.222 |
| Dreamlike | 2.570 | 2.421 | 2.509 | 1.760 | 1.659 | 1.958 | 2.413 | 2.376 | 2.482 | 1.446 | 1.471 | 1.645 | 2.194 |
| SD15 | 2.607 | 2.379 | 2.444 | 1.787 | 1.652 | 1.877 | 2.464 | 2.333 | 2.436 | 1.538 | 1.497 | 1.644 | 2.190 |
| SDXL | 2.436 | 2.330 | 2.484 | 1.606 | 1.610 | 1.862 | 2.333 | 2.275 | 2.442 | 1.480 | 1.524 | 1.698 | 2.129 |
| Animate | 2.293 | 2.213 | 2.392 | 1.743 | 1.703 | 2.129 | 2.223 | 2.210 | 2.334 | 1.519 | 1.600 | 1.757 | 2.094 |
| InstructPix | 2.082 | 2.207 | 2.190 | 1.854 | 1.579 | 1.679 | 2.249 | 2.388 | 2.432 | 1.204 | 1.240 | 1.227 | 1.989 |

4090 GPU. This moderate arithmetic power ensures running models successfully while avoiding overpowered arithmetic that would allow the LMM to easily outperform the traditional methods.

## 4.2 BENCHMARK RESULT AND DISCUSSION

Figure 5 shows the performance of 6 I2T models as encoders and 12 T2I models as decoders in image compression. For I2T, considering the different lengths of intermediate text, we show the bit-per-pixel (bpp) of each model together with the performance index, where ULB and ELB correspond to 0.024 and 0.0024 bpps respectively, namely 1,000 and 10,000 times from original RGB-8 images.

For 6 indicators in I2T LMMs, while GPT-4o (OpenAI, 2023) does not perform well on Text-FR, it significantly outperforms on Full-FR. This suggests that although its generated text carries limited information, it has a strong orthogonal relationship with the low-level details of the image. This semantic information effectively compensates for the information loss after image compression. In addition, the given text facilitates the subsequent T2I model in decoding high-quality images, and its performance across various NR indicators is also commendable. In comparison, MPlugOwl-2 (Ye et al., 2023) and InstructBLIP (Dai et al., 2023) can effectively encode images into text, but their results are still inferior to GPT-4o. The only viable competitor is ShareGPT (Chen et al., 2023b), but it has a bpp of around 0.008, which is significantly larger than the other 5 models. This data size exceeds ELB and occupies one-third of the available ULB space. Considering multiple factors, GPT-4o remains the most suitable I2T model as the CMC encoder.

For 8 indicators in T2I LMMs, 2 restorative models (Lin et al., 2024; Yang et al., 2024) exhibit overwhelming consistency in *Full* and *Image* modes with acceptable perception results, enabling faithful image reconstruction close to the ground truth. However, its applicability is limited for the other 2 modes, particularly under the strict ELB conditions. The performance of the remaining models falls into two distinct extremes, where RealVis (Civital, 2024) shows high consistency but PG25

Table 4: Benchmark result in *Pixel* and *Text* mode for 8 T2I generative models at extremely-low bitrate (10,000 times compression). [Key: Best; Second Best]

| Index | Pixel-FR ↑ | | | Pixel-NR ↑ | | | Text-FR ↑ | | | Text-NR ↑ | | | Overall ↑ |
|---|---|---|---|---|---|---|---|---|---|---|---|---|---|
| T2I Model | NSI | SCI | AIGI | NSI | SCI | AIGI | NSI | SCI | AIGI | NSI | SCI | AIGI | |
| PG25 | 1.789 | 1.641 | 1.779 | 3.542 | 3.425 | 3.939 | 1.798 | 1.634 | 1.762 | 3.646 | 3.628 | 3.944 | 2.386 |
| RealVis | 2.041 | 1.872 | 2.033 | 2.591 | 2.502 | 3.316 | 1.868 | 1.668 | 1.777 | 3.428 | 3.295 | 3.734 | 2.300 |
| PG20 | 1.901 | 1.812 | 1.948 | 2.472 | 2.325 | 3.338 | 1.772 | 1.619 | 1.745 | 3.675 | 3.617 | 3.963 | 2.274 |
| SSD-1B | 1.990 | 1.864 | 2.019 | 2.265 | 2.271 | 2.984 | 1.852 | 1.661 | 1.787 | 3.409 | 3.285 | 3.760 | 2.239 |
| Animate | 1.828 | 1.743 | 1.902 | 2.306 | 2.159 | 2.875 | 1.750 | 1.615 | 1.712 | 3.485 | 3.296 | 3.717 | 2.163 |
| Dreamlike | 1.986 | 1.877 | 1.991 | 2.195 | 2.129 | 2.623 | 1.779 | 1.620 | 1.705 | 3.233 | 2.917 | 3.302 | 2.132 |
| SDXL | 1.923 | 1.824 | 1.980 | 1.830 | 1.879 | 2.255 | 1.822 | 1.633 | 1.762 | 3.358 | 3.224 | 3.708 | 2.118 |
| SD15 | 2.000 | 1.856 | 1.951 | 2.165 | 1.948 | 2.314 | 1.760 | 1.609 | 1.654 | 2.683 | 2.364 | 2.498 | 1.988 |

Table 5: Benchmark result in *Full* and *Pixel* mode for 6 I2T models. [Key: Best; Second Best].

| Index | Full-FR ↑ | | | Full-NR ↑ | | | Pixel-FR ↑ | | | Pixel-NR ↑ | | | Overall ↑ |
|---|---|---|---|---|---|---|---|---|---|---|---|---|---|
| I2T Model | NSI | SCI | AIGI | NSI | SCI | AIGI | NSI | SCI | AIGI | NSI | SCI | AIGI | |
| GPT-4o | 2.617 | 2.475 | 2.584 | 1.914 | 1.808 | 2.445 | 2.041 | 1.872 | 2.033 | 2.591 | 2.502 | 3.316 | 2.439 |
| ShareGPT | 2.607 | 2.479 | 2.577 | 1.925 | 1.870 | 2.446 | 2.032 | 1.872 | 2.042 | 2.543 | 2.556 | 3.259 | 2.432 |
| Qwen | 2.592 | 2.473 | 2.581 | 1.894 | 1.799 | 2.353 | 2.034 | 1.890 | 2.036 | 2.531 | 2.364 | 3.176 | 2.396 |
| MPlugOwl-2 | 2.605 | 2.477 | 2.568 | 1.910 | 1.808 | 2.314 | 2.035 | 1.892 | 2.028 | 2.504 | 2.391 | 3.075 | 2.384 |
| LLAVA-1.5 | 2.599 | 2.465 | 2.565 | 1.880 | 1.799 | 2.276 | 2.025 | 1.876 | 2.028 | 2.498 | 2.420 | 3.041 | 2.381 |
| InstructBLIP | 2.589 | 2.473 | 2.571 | 1.842 | 1.736 | 2.192 | 2.027 | 1.882 | 2.035 | 2.424 | 2.339 | 2.961 | 2.346 |

(Li et al., 2024g) demonstrates high perception. Given that it is feasible to enhance a compressed low-quality image with high fidelity to the original, while correcting a completely different high-quality image with low fidelity remains challenging, we opt to prioritize consistency by assigning it a higher weight. Consequently, considering the strong performance and wide versatility of RealVis, it is relatively more suitable than the CMC decoder.

To delve into the compression capability of I2T and T2I LMMs with different content on various modes, we present the T2I leaderboard under ULB and ELB conditions in Table 3 and Table 4, respectively, and showcase the performance of I2T models on *Full* and *Pixel* modes (*Text* mode attached in appendix) in Table 5, with a discussion of content-specific analysis. A horizontal comparison among different modes in Tables 3 and 4 reveals that the *Full* mode has a clear advantage over the *Image* mode in terms of consistency and perception, indicating the significance of the text provided by the I2T model for T2I decoding. This text guidance not only enhances consistency but provides a clear target for the T2I process, thus also boosting perception. In contrast, the *Pixel* mode sacrifices perception for consistency compared to the *Text* mode. This is because the more control conditions added, the less room for creative freedom the model has, leading to a decrease in image aesthetics. However, for models that already have high perception scores (Li et al., 2024g; PlaygroundAI, 2023) in the *Text* mode, the trade-off of improving overall performance is acceptable.

Among NSI, SCI, and AIGI, different LMMs excel at different content. For instance, as shown in Table 3 and Table 4, PG25 (Li et al., 2024g), trained on internet data, performs better in AIGI tasks; conversely, RealVis aims at image naturalness, manifesting its superior reconstruction capability in NSI. Regardless of the model employed, we observe that NSI generally yields higher consistency scores, while AIGI has higher perception scores. However, SCI stands out from the others, with the compression results of the same model lagging behind in both perception and consistency. This deficiency is relevant to certain words (Shen & Zhao, 2014) (even long paragraphs) within SCI, making I2T models unable to re-encode them into text, while the text generation capabilities of recent T2I models are still limited. Besides, although the performance disparities among I2T models are not as significant as those in T2I models, Table 5 also clearly illustrates the limitation in SCI, indicating room for further optimization.

## 4.3 SUBJECTIVE DATA ANALYSIS

Figure 6 presents the subjective preference for images decoded by 12 T2I models under ULB and 8 models under ELB. For ULB, the 3 restorative models (Lin et al., 2024; Yang et al., 2024; Wang et al., 2023b) exhibit slightly higher consistency compared to generative models, where PG25 achieves the highest perception score against all others. It is worth noting that the restorative mod-

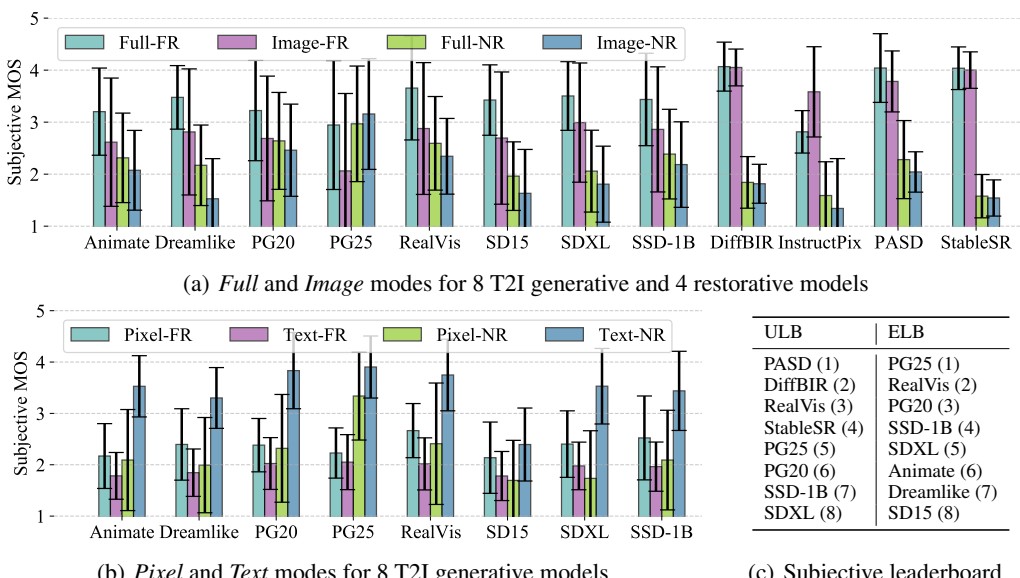

(a) *Full* and *Image* modes for 8 T2I generative and 4 restorative models

(b) *Pixel* and *Text* modes for 8 T2I generative models

(c) Subjective leaderboard

Figure 6: Illustration of subjective preference in terms of Mean Opinion Score.

Table 6: Complexity analysis of CMC and traditional VVC codec on 1024*1024 image. Using LMM for compression consumes more decoding resources but remarkably saves encoding time.

| Encoding/Decoding | Animate | Dreamlike | PG20 | RealVis | VVC | GPT-4o | InstructBlip | LLaVA | MPlugOwl | VVC |
|---|---|---|---|---|---|---|---|---|---|---|
| Time (s) | 6.48 | 1.62 | 6.30 | 5.48 | 83.88 | 12.84 | 2.14 | 2.46 | 4.26 | 0.194 |

els are more robust. The upper and lower bounds of the scores in each dimension seldom surpass 1.0, whereas the randomness of the generative models notably deteriorates. As the bitrate further decreases to ELB, consistency scores of all models decline, while perception scores have slight improvement. In summary, apart from Animate (Guo et al., 2024) specifically for cartoon styles, and InstructPix (Brooks et al., 2023) that significantly alters images, all other models demonstrate potential applications in CMC. Additionally, by averaging all scores, we find that the models ranking based on subjective scores aligns closely with the objective ones shown in Table 3 and Table 4. This finding validates the reasonability of our previous experiments and highlights that, compared to perception, humans tend to focus more on consistency when viewing compressed images.

## 4.4 COMPARE WITH TRADITIONAL CODECS

To validate the practicality of the CMC paradigm, we select 2 outstanding combinations of I2T and T2I models from CMC-Bench, and compare them with 3 mainstream codecs: AVC (Wiegand et al., 2003), HEVC (Sullivan et al., 2012), and VVC (Bross et al., 2021) at I-Frame mode, and the latest semantic codec pipeline CDC (Yang & Mandt, 2023). Given the superiority of GPT-4o (OpenAI, 2023) as the encoder, we initially pair it with the top-ranked decoder DiffBIR (Lin et al., 2024). Considering applications on different modes, excluding the reconstructive model, we also assess its performance with the third-tanked decoder RealVis (Civital, 2024). These two approaches with four bitrates correspond to *Text*, *Pixel*, *Image*, *Full* modes are shown in Figure 7. To comprehensively compare the two paradigms across different dimensions, we add 3 Consistency metrics: CLIPSIM (Radford et al., 2021), LPIPS (Zhang et al., 2018a), and SSIM (Wang, 2004); and 3 Perception metrics: CLIPIQA (Wang et al., 2023a), LIQE (Zhang et al., 2023b), and FID (Heusel et al., 2017). Models ranked higher prioritize semantic information, while those lower focus on pixel-level consistency. First, we compared the execution speeds, listed in Table 6.Here, we set the QP of VVC to 41, in which case its performance is roughly similar to that of CMC, thus ensuring a fair comparison. It can be seen that no matter what combination of LMMs is used, CMC is more suitable for encoding and the conventional method is faster for decoding. Thus, CMC already has a certain application value at present and has the potential to replace VVC in the future. After that, we analyze it in terms of FR/NR metrics.

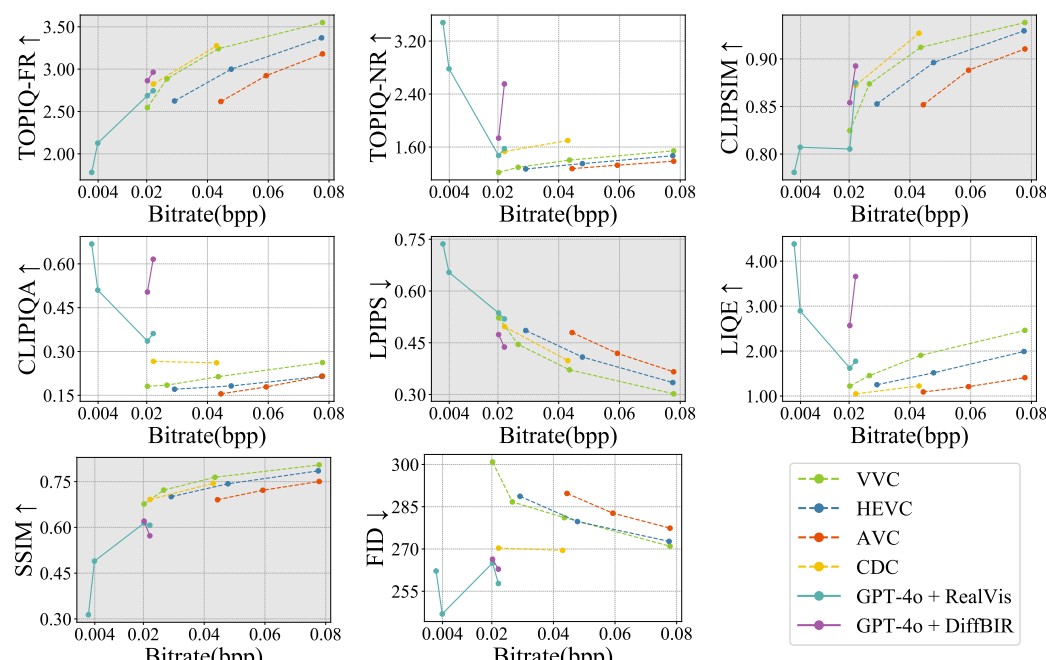

Figure 7: Comparison of CMC-Bench winners against existing image codecs, evaluated by 4 consistency and 4 perception metrics indicated by marked and plain background. The combination of I2T and T2I models generally exceeds the existing codecs under the same bitrate.

Both CMC paradigms demonstrate an advance in terms of most metrics. Given that SSIM is purely pixel-based, the performance drop due to generative compression is expected. The lead in perception is particularly notable, as it surpasses traditional codecs at extremely low bitrates. However, the advantage in consistency is relatively smaller, achieving a reduction of around 30% in bitrate compared to traditional methods at 0.02 bpp. The DiffBIR generally shows better performance, while RealVis fits A wider range of bitrates. In summary, based on the above analyses, we believe that CMC holds a certain advantage over traditional encoding. However, for implementing LMMs into the next generation of visual signal codecs, further optimization is required in the following aspects:

**Enhanced T2I models**: Both encoders and decoders are crucial in CMC, but decoders are more decisive. Future T2I models should possess more sophisticated control mechanisms, ensuring high-quality generation while maintaining consistency with reference images and text.
**Better adaption to SCI**: the compression performance of SCI is inferior to NSI and AIGI, necessitating LMMs with specialized understanding and generating mechanisms to handle SCI.
**Wider bitrate range**: Although leading in ULB and ELB, the margin of consistency improvement is not as pronounced as perception. Future efforts should focus on CMC at higher bitrates, incorporating more control information to aid in reconstructing the original image, ultimately achieving superiority across all bitrates and dimensions as compared to traditional codecs.

## 5  CONCLUSION

We construct CMC-Bench, a benchmark for assessing the collaborative functioning of I2T and T2I models in image compression. Anticipating the bitrate requirements for codecs in the next decade, we proposed four collaboration modes among LMMs, along with two indicators of consistency and perception. By employing 6 mainstream I2T and 12 T2I models, we collected 58,000 distorted images through CMC with 160,000 human subjective annotations to train objective metrics for comprehensive evaluation. Our assessment demonstrates that even without dedicated training for compression tasks, combinations of several advanced I2T and T2I models have already surpassed traditional codecs in multiple aspects. However, there is still a long way to go before LMMs can directly become the future codecs paradigm. We sincerely hope that CMC-Bench will inspire future LMMs to perform better compression towards the evolution of visual signal codecs.

## ETHIC STATEMENT

The research conducted in the paper conforms, in every respect, with the ICLR Code of Ethics. The data collection, processing, and analysis all comply with the declaration of Helsinki. Official ethical certificates and stamps of approval were obtained before the experiment. Each user provides informed consent for their data to be used in experiments. as shown in Figure A1.

You are being asked to participate in a research study. Before you decide, it is important for you to understand why the research is being done and what it will involve. Please take your time to read the following information carefully and ask questions about anything you do not understand. This form describes a research study that you are invited to take part in.

Purpose of the Study The purpose of this study is to annotate your preference towards AI-Generated Images

Procedures If you agree to take part in this study, the researcher will collect and use data from your preference. The data may include, but is not limited to, scientific research, subjective analysis, model training.

Risks There are minimal foreseeable risks associated with the use of your data for research purposes. However, as with any data collection and storage, there is a risk of unauthorized access despite all reasonable security measures being taken.

Benefits The potential benefits of this research include 40-80 CNY according to your annotation quality.

Confidentiality Your data will be treated confidentially and will only be accessible to the researcher(s) involved in this study. All identifiable personal information will be kept confidential and will not be shared outside of the research team.

Voluntary Participation and Withdrawal Your participation in this study is voluntary. You have the right to refuse to participate or to withdraw your consent at any time without affecting your current or future relations with the researcher or organization.

I have read the above information, and I have had the opportunity to ask questions and have had those questions answered to my satisfaction. By providing my data and signing below, I consent to participate in this research study and for my data to be used for research purposes.

Please Enter your Name in English

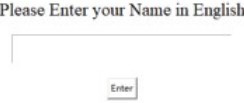

Figure A1: Data Collection Agreement.

## REPRODUCIBILITY STATEMENT

We have provided implementation details in Sections 4.1 and the Appendix. We will also release all the code. The benchmark is a long-term project, which will be updated every month by the CMC-Bench author team. We look forward to testing the 308 effectiveness of more advanced LMMs on CMC tasks in the future. All users are free to use R-Bench-related resources, except subject's personal preferences will be protected. If anyone wants to extend the benchmark, including but not limited to I2T+T2I pipeline, only T2I/I2T models, and different data content beyond NSI/SCI/AIGI can contact us and their contributions will be reviewed.

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

## A  APPENDIX

In this section, we briefly describe the content of the checklist requirements. Considering that our experiments tried a variety of parameter configurations, the conclusions under different configurations are also stated here, including specific ablation data.

## A.1 LIMITATIONS AND SOCIAL IMPACT

**Limitation 1**: Although we have considered most of the mainstream I2T and T2I models in CMC-Bench (till March 2024), the number of models is still insufficient to fully characterize the performance of all current LMMs on CMC. Taking the open-source T2 model as an example, more than 20,000 models have been released on huggingface (till May 2024). Although we cannot run all models, the capabilities of some relatively unpopular or more advanced LMMs in the future need to be further updated on CMC-Bench.

**Limitation 2**: CMC-Bench is currently designed for the performance verification of image compression, not video compression. Considering that the temporal information of videos is relatively complex, the current LMMs are only applicable to image compression, which makes it difficult to ensure consistency with the reference when generating videos. However, as LMMs gradually apply to video compression in the future, CMC-Bench will also be evaluated at the video level.

**Social Impact**: Through the CMC paradigm, the size of the image can be compressed by 1,000 times, and even 10,000 times in extreme cases. This will effectively promote image communication between a large number of terminals under limited bandwidth, thereby realizing multi-device collaboration in the Internet of Things and semantic communication. Considering that traditional codecs have encountered bottlenecks after three decades of development and the compression rate is gradually approaching the Shannon limit, we believe that LMM will effectively achieve semantic-level compression and thus become the future evolution direction of visual information codec protocol.

From an industry perspective, CMC is ready for the following two real-world scenarios. (1) Communication: In severe scenarios, channel resources are extremely limited, such as deep sea and space; or there are too many devices, that is, hundreds of devices in IoT share a local area network. At this time, traditional compression methods cannot adapt to such a low bit rate and can only communicate images through CMC; (2) Storage: According to statistics from mainstream social media, 10% of visual information contributes to 99% of views, and most images are 'junk data'. For these images that are rarely clicked but not suitable for deletion, their storage will consume considerable resources. Therefore, they can be compressed in CMC format and decompressed when needed. With the advancement of LMM, models with lower complexity have emerged in recent years. We believe CMC can move from these two applications without latency requirements to real-time scenarios.

## A.2 SUBJECTIVE ANNOTATION SETTINGS

Compliant with the ITU-R BT.500-13 (Union, 2002) standard, we invited 20 viewers (11 male, 9 female) in this subjective experiment with normal lighting levels. Images are presented on the iMac display together with the ground truth in random order on the screen, with a resolution of up to $4096 \times 2304$. Both ground truth and distorted images are accessible for subjective. Considering the consistency between the reference and distorted image, and the perceptual quality of the only distorted image, subjects were asked to give two scores within the range of [0, 5], where each one-point interval stands for poor, bad, fair, good, or excellent quality. The user interface is shown in Figure A2.

Each user, in accordance with the Helsinki Declaration, provides informed consent for their data to be used in experiments. To prevent NSFW content, we implement three preventive measures: (1) Conduct a thorough manual screening of the ground truth; (2) Utilize the SD safety checker (Rombach et al., 2022a) during decoding; (3) Incorporate an 'offensive' flag in the annotation process, allowing viewers to report NSFW content if encountered. The data confirms that the ground truth is safe, with approximately 0.2% of distorted images receiving reports, which is generally acceptable.

In case of visual fatigue, we split the database into $g \in [0, 10]$ groups including $M = 400$ images each, while limiting the experiment time to an hour. After collecting every viewer's quality ratings, we compute the Spearman Rank-order Correlation Coefficient (SRoCC) between them and the global average and remove the outliers with SRoCC lower than 0.6. Then we normalize the average score $s$ for between each session to avoid inter-session scoring differences as:

$$s_{ij}(g) = r_{ij}(g) - \frac{1}{M} \sum_{i=0}^{g \cdot M - 1} r_{ij} + 2.5, \tag{1}$$

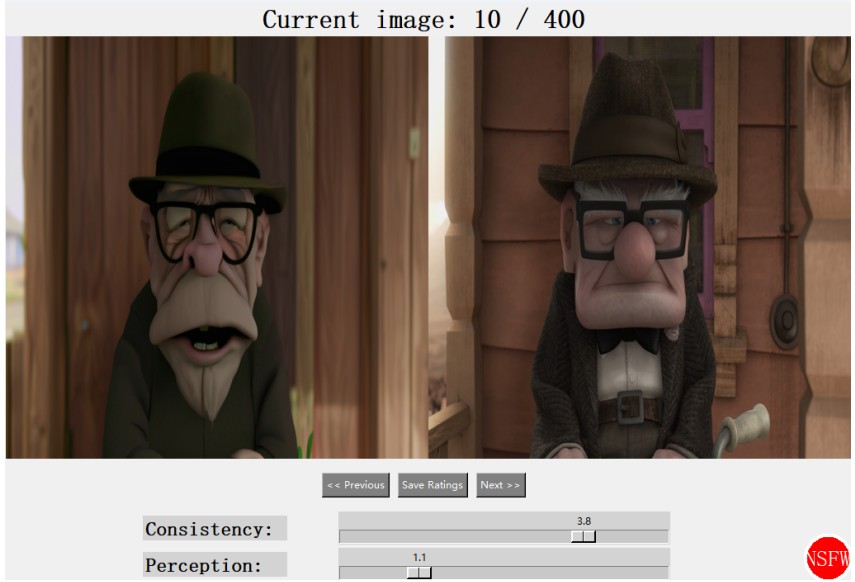

Figure A2: Subjective annotation interface presenting with the distorted (left) and reference (right) image. Each viewer is asked to provide (1) a Consistency score between two images from 0 to 5; (2) a Perception score of the distorted image; (3) an NSFW flag when they feel offended.

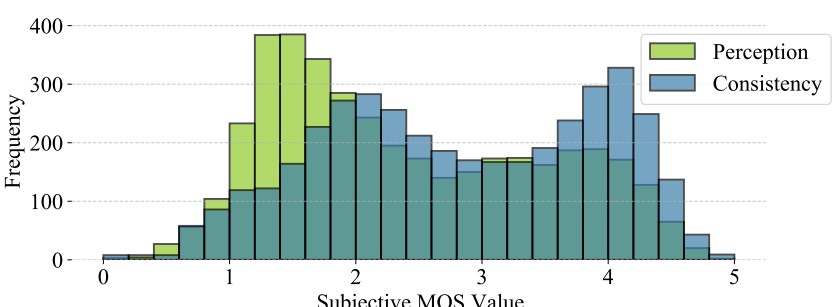

Figure A3: Mean Opinion Score distribution of the Consistency and the Perception dimension.

where $(i, j)$ represent the index of the image and viewer and $r$ stands for raw score. We observed a fairly even distribution of subjective scores on both dimensions and bar graphs for each score range are provided in the appendix. Then subjective scores are converted to Z-scores $z_{ij}$ by:

$$z_{ij} = \frac{s_{ij} - \mu_j}{\sigma_j}, \tag{2}$$

where $\mu_j = \frac{1}{N} \sum_{i=0}^{N-1} s_{ij}$, $\sigma_j = \sqrt{\frac{1}{N-1} \sum_{i=0}^{N-1} (s_{ij} - \mu_i)^2}$ and $N = 10$ is the number of subjects, which is finally reported as MOS, namely golden user annotations. The distribution of Consistency and the Perception MOS is shown in Figure A3, which proves that extremely low and high scores are rare, and most scores are between 1 and 4. The Perception score is concentrated in the medium-low area, while the Consistency dimension tends to be moderately high.

A.3    EXPERIMENTAL PLATFORM

For 4,000 labeled image pairs, we trained five FR and five NR quality indicators for 50 epochs using Adam optimizer on a local NVIDIA GeForce RTX 4090. Among which 80/20 for training/testing. We take MSE loss with a learning rate at $2 \times 10^{-5}$. The TOPIQ-FR and TOPIQ-NR are set as objective indicators for Consistency/Perception. Noted these 4,000 training data images are not included in the source data for objective evaluation for a fair comparison.

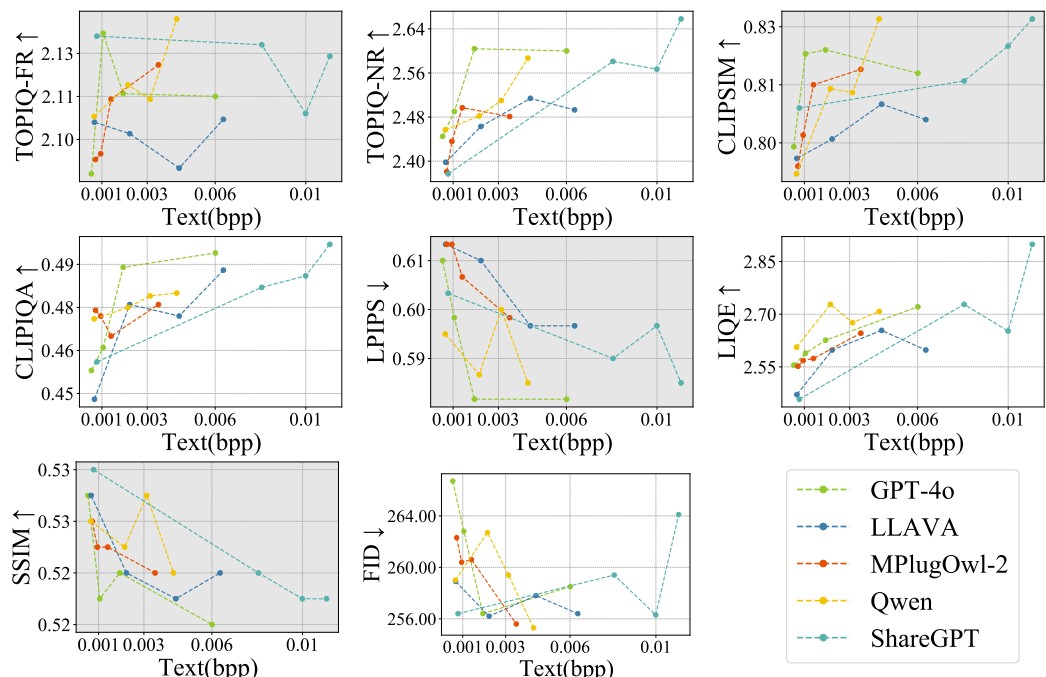

Figure A4: Setting I2T model with different output lengths for CMC task in *Full* mode, evaluated by 4 consistency and 4 perception metrics indicated by marked and plain background. A moderate output length can realize a satisfying performance while saving bitrate.

The LMMs are validated on a server with four NVIDIA RTX A6000, using I2T in different output lengths, and T2I in different strengths, combining through *Text/Pixel/Image/Full* modes.

### A.4 IMAGE-TO-TEXT MODEL CONFIGURATION

Towards different output lengths, we applied different prompts as the input for I2T models.

To explore how much information the above four output lengths can represent. We use different I2T models, and combine them with the most effective T2I model (RealVis (Civital, 2024)) under the above four output lengths, and use four Consistency and four Perception indicators for analysis, as shown in Figure A4, where the four datapoints of each curve represent four output lengths. The experimental results show that most I2T models can dynamically adjust the output length, except that InstructBLIP for image annotation cannot input prompt, and ShareGPT is not sensitive to the specified output length. Overall, when inputting Q(5), the reconstruction effect is relatively poor because of the short output; when inputting Q(50), the overly long paragraph from the I2T model cannot be understood by the T2I model, so the performance is not significantly improved while wasting bitrate. By observing the trend of all curves, we find that when the bpp of the text is between 0.002-0.003, the balance between performance and bitrate can be achieved. Therefore, for each model, we choose the output length closest to this bitrate, that is, Q(20) for GPT-4o (OpenAI, 2023), MPlugOwl-2 (Ye et al., 2023); Q(10) for LLAVA (Liu et al., 2023), Qwen (Bai et al., 2023), ShareGPT (Chen et al., 2023b); and the default length for InstructBLIP (Dai et al., 2023).

### A.5 TEXT-TO-IMAGE MODEL CONFIGURATION

The verification of *Full/Image/Pixel* results are shown in Figure A5/A6/A7 respectively, using same 4 Consistency and 4 Perception indicators. In general, as the strength increases, the Consistency index increases first and then decreases, while the Perception index continues to rise. This is because the greater the strength, the more details the T2I model adds to the image, thereby improving the Perception score. However, for Consistency, the added details at low strength can indeed make up for the unclear areas in the reference image, thereby performing restoration; but when the strength increases, the added details are inconsistent with the original image, and instead bring negative

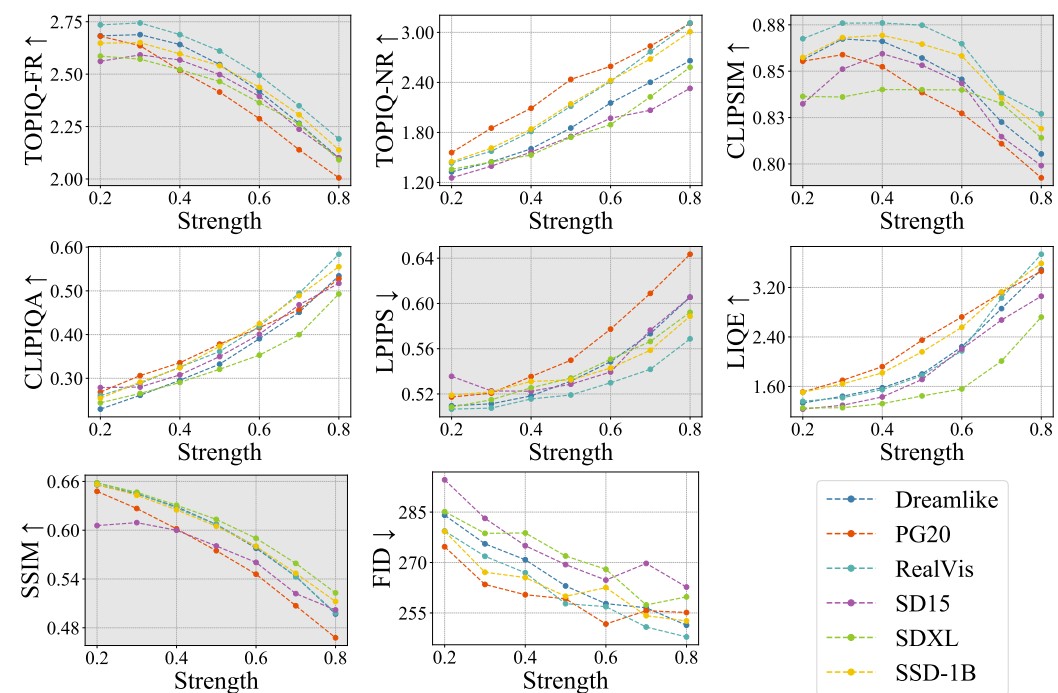

Figure A5: Setting T2I model with different strength for CMC task in *Full* mode, evaluated by 4 consistency and 4 perception metrics indicated by  marked  and plain background. A strength of 0.5 can reach a balance between consistency and perception.

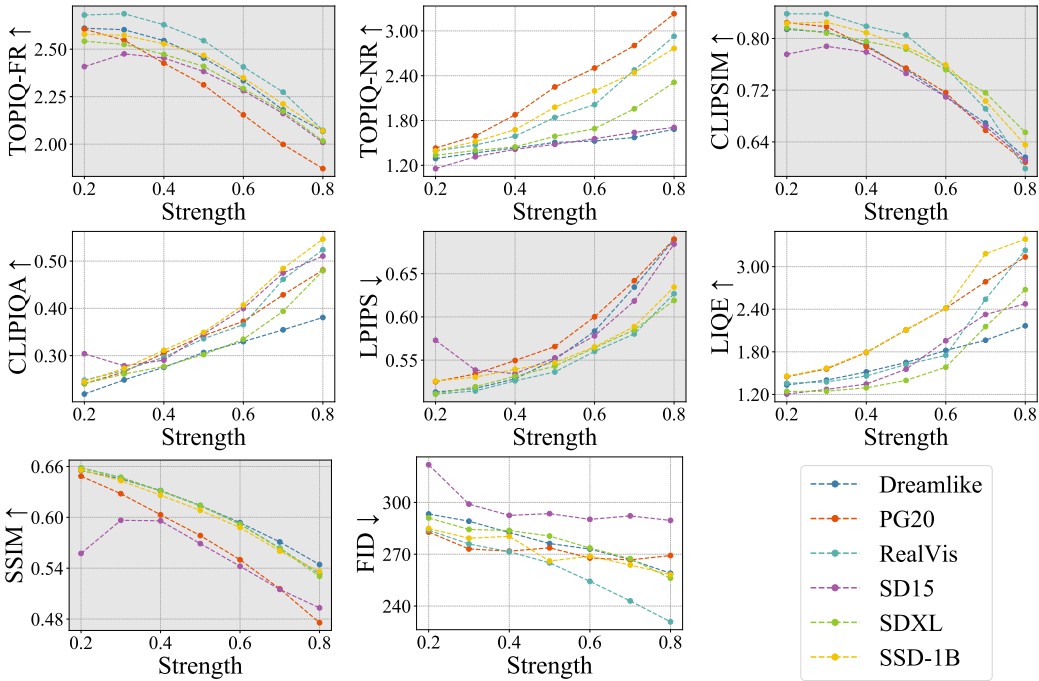

Figure A6: Setting T2I model with different strength for CMC task in *Image* mode, evaluated by 4 consistency and 4 perception metrics indicated by  marked  and plain background. A strength of 0.5 can reach a balance between consistency and perception.

optimization to the reference image. Thus, a good strength requires a trade-off between Consistency and Perception. Taking both dimensions into consideration, we set the strength of *Full* and *Image* mode to 0.5 and the *Pixel* mode to 0.8.

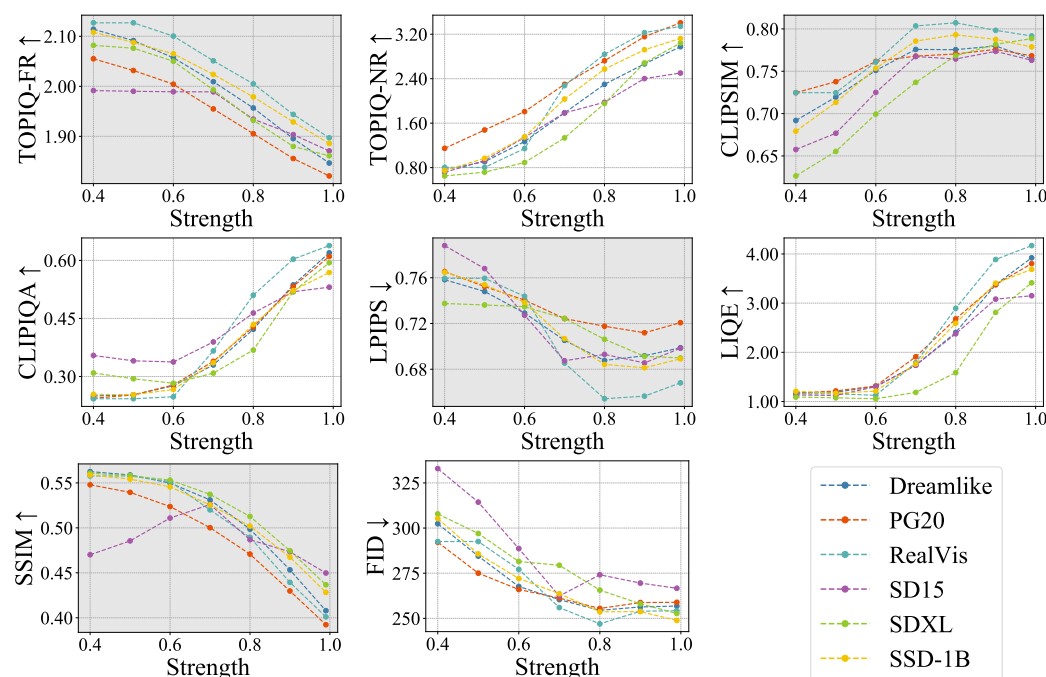

Figure A7: Setting T2I model with different strength for CMC task in *Pixel* mode, evaluated by 4 consistency and 4 perception metrics indicated by `marked` and plain background. A strength of 0.8 can reach a balance between consistency and perception.

Table A1: Changing reference image as different VVC QP level in *Full* mode before CMC decoding, evaluated by 4 consistency and 4 perception metrics indicated by `marked` and plain background.

| Reference | Image | CLIPSIM ↑ | CLIPIQA ↑ | LPIPS ↓ | LIQE ↑ | SSIM ↑ | FID ↓ | TOPIQ-FR ↑ | TOPIQ-NR ↑ |
|---|---|---|---|---|---|---|---|---|---|
| Extreme (*Full* mode) | Original | 0.825 | 0.181 | 0.523 | 1.221 | 0.677 | 300.8 | 2.546 | 1.217 |
| | CMC | 0.890 | 0.670 | 0.435 | 3.530 | 0.571 | 255.8 | 2.964 | 2.553 |
| | Improve | 7.88% | 270% | 20.2% | 189% | -15.6% | 17.5% | 16.4% | 109% |
| QP51 | Original | 0.874 | 0.185 | 0.445 | 1.454 | 0.722 | 286.7 | 2.886 | 1.294 |
| | CMC | 0.919 | 0.696 | 0.383 | 3.749 | 0.586 | 255.2 | 3.202 | 2.810 |
| | Improve | 5.15% | 276% | 16.1% | 157% | -18.8% | 12.3% | 10.9% | 117% |
| QP48 | Original | 0.912 | 0.214 | 0.372 | 1.905 | 0.764 | 281.1 | 3.241 | 1.403 |
| | CMC | 0.931 | 0.704 | 0.350 | 3.884 | 0.603 | 254.2 | 3.342 | 2.981 |
| | Improve | 2.08% | 228% | 6.29% | 103% | -21.0% | 10.5% | 3.12% | 112% |
| QP45 | Original | 0.938 | 0.262 | 0.302 | 2.461 | 0.805 | 270.9 | 3.550 | 1.542 |
| | CMC | 0.939 | 0.711 | 0.327 | 3.999 | 0.611 | 254.0 | 3.431 | 3.052 |
| | Improve | 0.11% | 171% | -7.65% | 62.4% | -24.1% | 6.67% | -3.35% | 97.9% |

## A.6    APPLICABILITY ON DIFFERENT REFERENCE IMAGE

In the main text, the VVC provides the reference image with QP=53. In Table A1 we further use VVC with QP=51,48,45 as the reference image for T2I model denoising to perform CMC. At higher bitrates, CMC still has an overwhelming advantage over VVC in the Perception metric. Except for SSIM, CMC achieves comprehensive optimization of all other indicators compared to traditional codecs, but the optimization range gradually decreases with the increase of bitrate. Moreover, once the QP is lower than 45, it will fall behind in the Consistency indicators. In summary, compared with traditional codecs, CMC can achieve an overall improvement in Perception and Consistency at low bitrates. However, when bpp increases to 0.1 or above, the improvement in Perception comes at the cost of Consistency. This indicates that ideal performance at higher bitrates is an important factor when using LMMs for image compression.

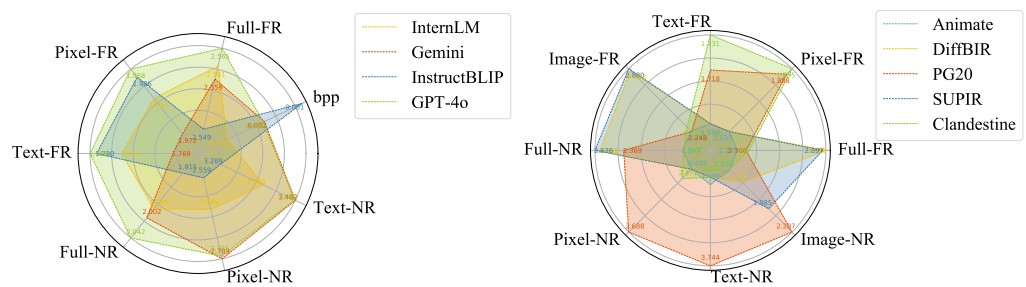

Figure A8: A radar map illustrates the collaboration of mainstream I2T (left) and T2I (right) LMMs. The model are tested as {4 different I2Ts + RealVis} and {GPT-4o + 5 different T2Is}.

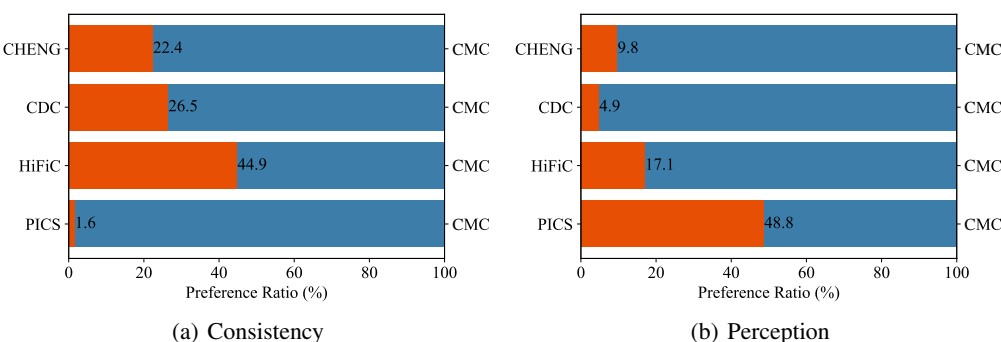

(a) Consistency                    (b) Perception

Figure A9: User subjective preference between CMC and three advance codecs, where CMC is satisfying in both consistency and perception level.

## A.7 FURTHER EXPERIMENTAL ANALYSIS

### A.7.1 T2I STRUCTURE

The CMC-Bench mainly discussed the diffusion structure for T2I models instead of Auto-Regressive (AR). This is not because the AR performance metrics are insufficient, but because generating pixels one by one leads to a higher complexity. We acknowledge that the AR model is more controllable and may sometimes achieve better performance than the Diffusion model, but it requires a massive amount of parameters for the T2I model and longer decoding times. Work Sun et al. (2024) shows AR generating a 384*384 image takes more than 6 seconds, which is enough time to generate several 1024 resolution images with a Diffusion model whose size is less than 1B. Considering the real-time nature of visual communication, we believe that the Diffusion model is more suitable for CMC.

Among the Diffusion architectures, their performance requires a case-by-case discussion, including the Restoration, SDXL, and SD architectures. The leaderboard in Figure A3 shows that on the ULB, the order is Restoration > SD > SDXL, because Restoration fine-tunes based on the original image, SD uses the original image as the starting point to draw, and SDXL, compared to SD, has more freedom to make changes, which can lead to excessive modifications. On the ELB, the order is SDXL > SD, because without a reference image, Restoration is directly unavailable. Compared to SDXL's aggressive strategy, which becomes an advantage, it can produce more details.

### A.7.2 MODE ADVANCED I2T AND T2I MODELS

We considered some more advanced T2I models, including two I2T (Gemini, InternLM-XComposer2) and two T2I (Clandestine, SUPIR). Their performance is shown in Figure A8. It can be seen that these models have not yet shaken the leading position of GPT4-o as I2T and DiffBIR as T2I. We welcome more I2T/T2I developers to participate in the test.

### A.7.3 THE CONSISTENCY-PERCEPTION TRADE-OFF

The consistency and perception have always been a dilemma in the image compression task. For low-bitrate image compression ($< 0.1$ bpp) Blau & Michaeli (2018), the compression algorithm provides a rough encoding of the original image, necessitating the decoder to add details. Inadequate detail leads to poor perceptual quality, while excessive detail results in inconsistency with the original image. As bitrates decrease further to ultra-low levels ($< 0.024$ bpp) Blau & Michaeli (2019), the conflict between these two objectives becomes even more intensified.

Therefore, ensuring both consistency and perceptual quality at such low bitrates remains a challenge. When using LMM for compression, adding how many details, is the decisive factor for such a trade-off. Thus, we believe the CMC-Bench enables LMM developers with comprehensive metrics that cater to both objectives.

### A.7.4 USER STUDY

To verify the practicality of CMC in real-life scenarios, we conduct a subjective user study beyond the objective indicators, to analyze the human preference for the compressed image. We established an environment with standard lighting, displaying the ground truth centrally, and two compressed images on a monitor with a × 2,304 resolution. Viewers are required to select preferences between two images compressed by different algorithms, at both consistency and perception levels. The experiment involved 5 graduate students (2 males and 3 females) as subjects. CMC, using the Full mode of GPT-4o+DiffBIR, is compared with four state-of-art compression metrics, namely CHENG Cheng et al. (2020), CDC Yang & Mandt (2023), HiFiC Mentzer et al. (2020), and PICS Lei et al. (2023). Same bitrate is set for all those metrics for a fair comparison. The validation results illustrated in Figure A9 demonstrate the superior performance of MISC across all evaluated criteria. Notably, CMC performs comparably to the PICS for consistency, and HiFiC for perception. Furthermore, compared to NSIs, AIGIs compressed by MISC were more preferred by human evaluators.

### A.8 EXAMPLE RESULT VISUALIZATION

The CMC result visualization is shown from Figure A11 to A16, all result use GPT-4o (OpenAI, 2023) as encoder, and Animate(Guo et al., 2024)/ Dreamlike(dreamlike art, 2023)/PG20(PlaygroundAI, 2023)/PG25(Li et al., 2024g)/RealVis(Civital, 2024) as decoder (from left to right). Four working modes *Full*/*Image*/*Pixel*/*Text* are all included (from top to bottom). For different modes, the compression results from LMMs show that as the bitrate decreases, the decoded image is more different from the ground truth. Among them, the *Full* mode can obtain results generally similar to the ground truth; the *Image* mode will lose some semantic details while introducing artifacts; the *Pixel* mode loses more details but ensures the consistency of the overall composition; and the result generated by *Text* is significantly different from the ground truth.

For the performance of the CMC on different contents, Figure A11/A12 reveals it performs most satisfactorily on AIGIs; Figure A13/A14 indicates it can also obtain results consistent with ground truth on NSIs, but it is easy to lose details such as human faces and vehicle signs; Figure A15/A16 implies it is the least ideal on SCIs, as it misunderstands the relationship between characters in the movie or games, and cannot draw formed letters on webpages. In conclusion, CMC is a promising visual signal compression method, but to become a universal codec standard in the future, the robustness to all content types needs to be improved.

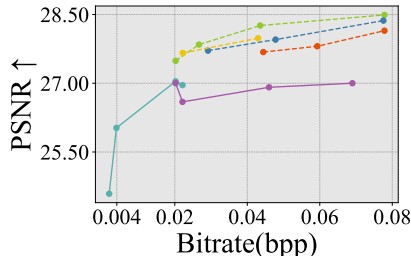

Figure A10: Added PSNR result, color legends same as Figure 7.

### A.9 DATA STATEMENT

The CMC-Bench dataset is released under the **CC BY 4.0** license. This includes all ground truth, distorted images, subjective annotations, and the weight of the Consistency/Perception evaluation model. All LMM developers can test their performance through our public scripts, and all image compression researchers can obtain the public I2T+T2I LMM pipeline. We believe these resources can inspire the next generation of visual signal codec protocols.

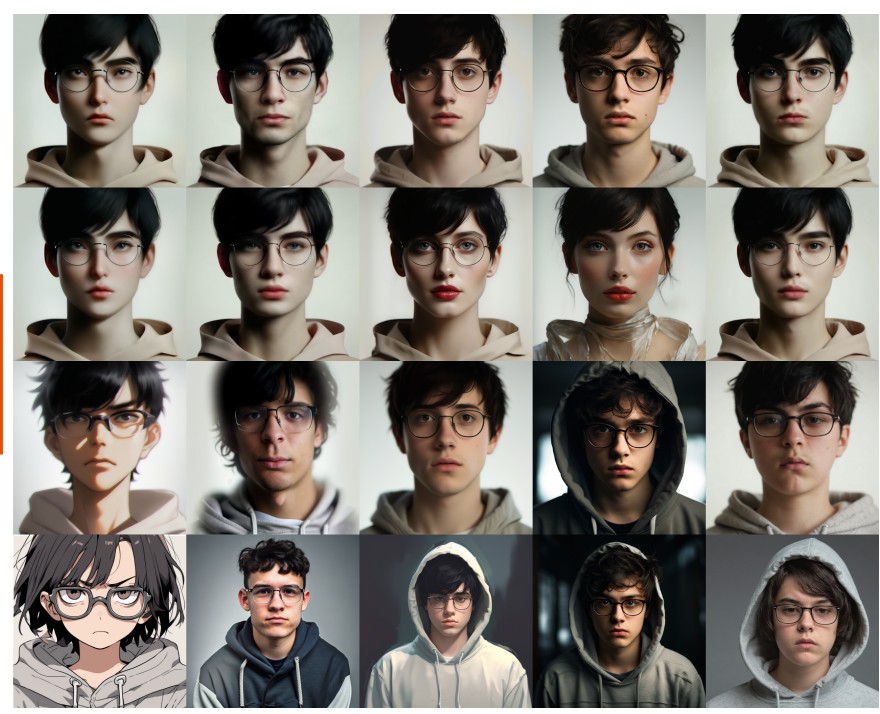

Figure A11: Visualization of an AIGI (Human) on the left after CMC. Row: *Full/Image/Pixel/Text* mode. Column: Animate(Guo et al., 2024)/ Dreamlike(dreamlike art, 2023)/PG20(PlaygroundAI, 2023)/PG25(Li et al., 2024g)/RealVis(Civital, 2024) as decoder.

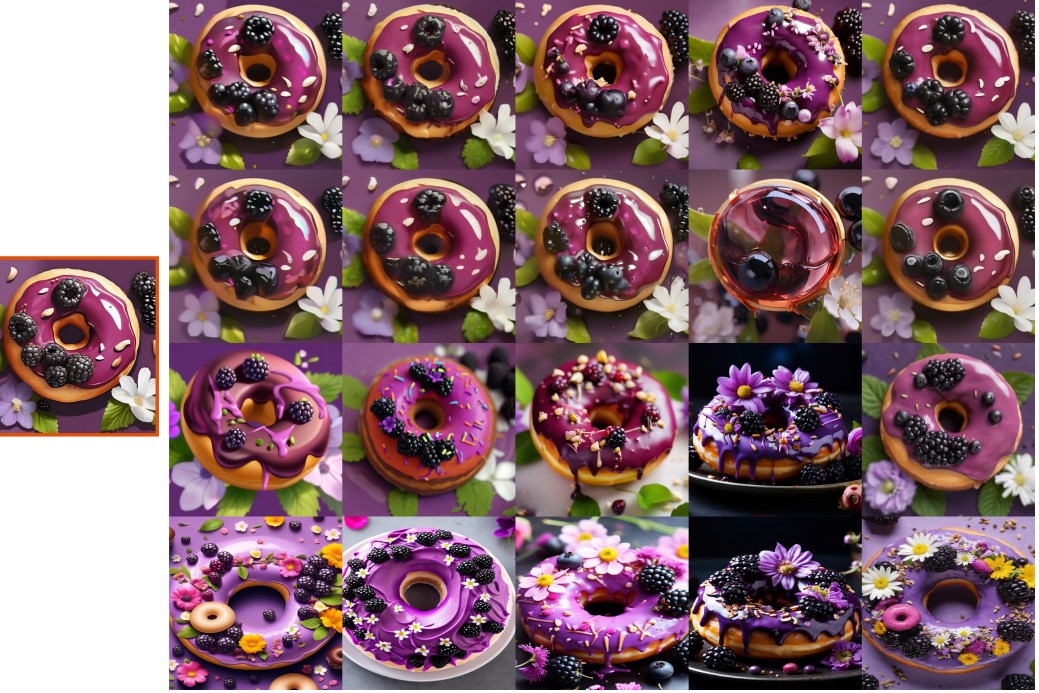

Figure A12: Visualization of an AIGI (Object) on the left after CMC. Row: *Full/Image/Pixel/Text* mode. Column: Animate(Guo et al., 2024)/ Dreamlike(dreamlike art, 2023)/PG20(PlaygroundAI, 2023)/PG25(Li et al., 2024g)/RealVis(Civital, 2024) as decoder.

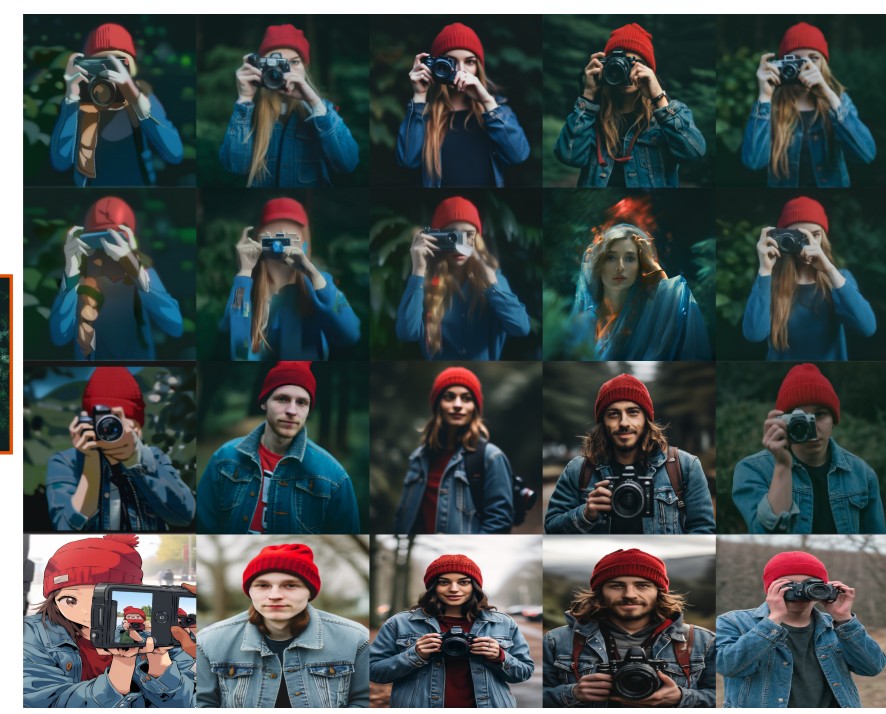

Figure A13: Visualization of an NSI (PGC) on the left after CMC. Row: *Full/Image/Pixel/Text* mode. Column: Animate(Guo et al., 2024)/ Dreamlike(dreamlike art, 2023)/PG20(PlaygroundAI, 2023)/PG25(Li et al., 2024g)/RealVis(Civital, 2024) as decoder.

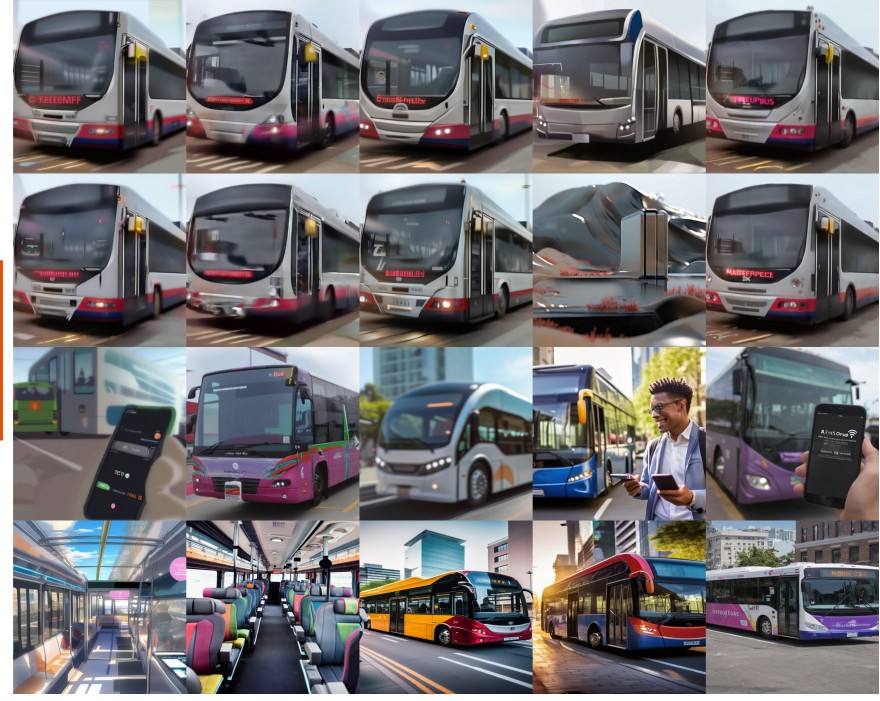

Figure A14: Visualization of an NGI (UGC) on the left after CMC. Row: *Full/Image/Pixel/Text* mode. Column: Animate(Guo et al., 2024)/ Dreamlike(dreamlike art, 2023)/PG20(PlaygroundAI, 2023)/PG25(Li et al., 2024g)/RealVis(Civital, 2024) as decoder.

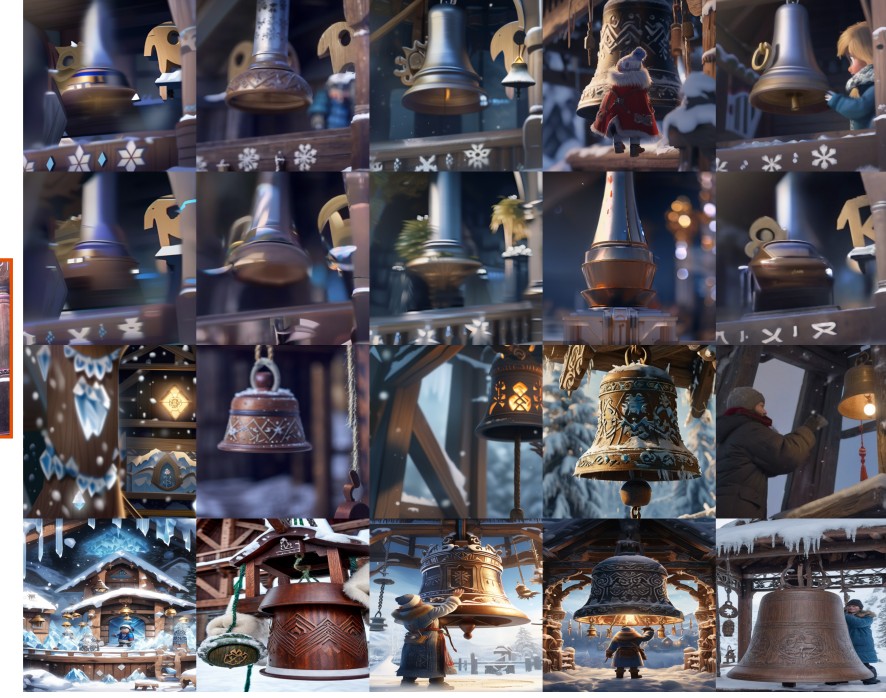

Figure A15: Visualization of an SCI (Movie) on the left after CMC. Row: *Full*/*Image*/*Pixel*/*Text* mode. Column: Animate(Guo et al., 2024)/ Dreamlike(dreamlike art, 2023)/PG20(PlaygroundAI, 2023)/PG25(Li et al., 2024g)/RealVis(Civital, 2024) as decoder.

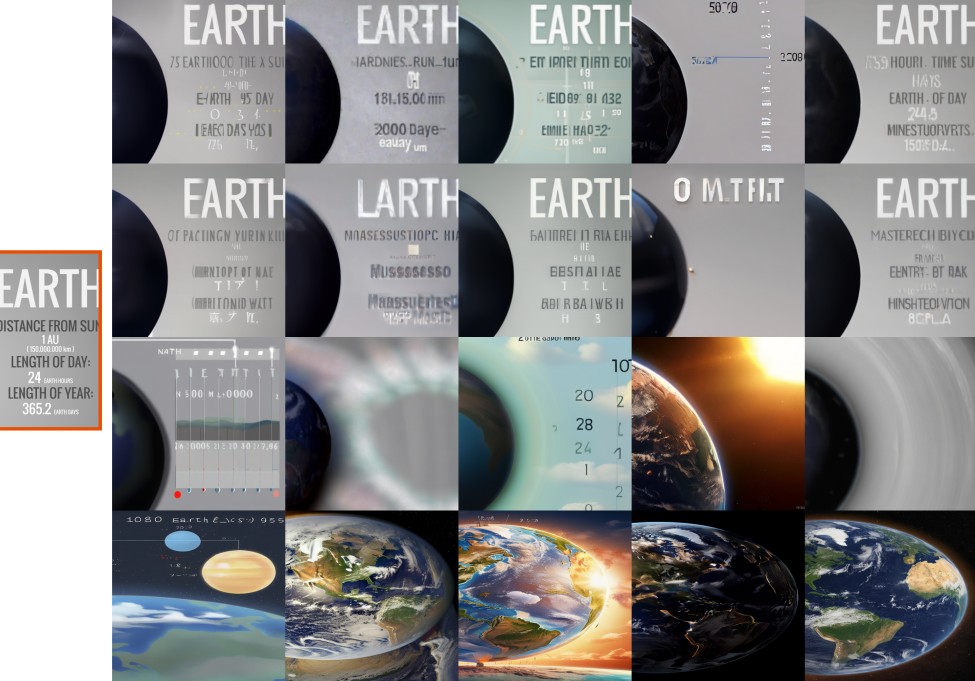

Figure A16: Visualization of an SCI (Webpage) on the left after CMC. Row: *Full*/*Image*/*Pixel*/*Text* mode. Column: Animate(Guo et al., 2024)/ Dreamlike(dreamlike art, 2023)/PG20(PlaygroundAI, 2023)/PG25(Li et al., 2024g)/RealVis(Civital, 2024) as decoder.

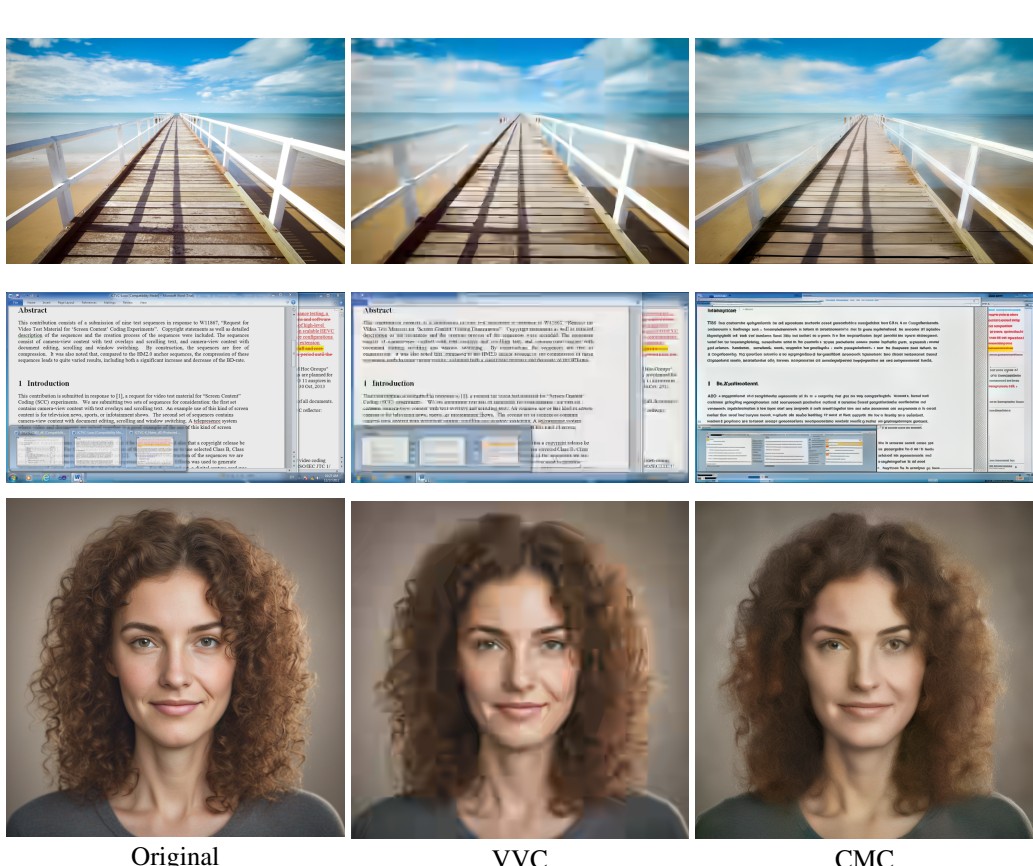

Figure A17: Success and failure cases for CMC. Left is the original image, the middle is compressed by VVC, right is compressed by CMC (GPT-4o+RealVis). NSI/SCI/AIGI cases are shown from up to bottom. The bitrate is fixed at about 0.024bpps. We found CMC is more successful than the advanced VVC, in the upper NSI and bottom AIGI; however, for the middle SCI, both of them cannot draw the words on the website, but at least CMC is not blurry. Zoom in for details.

