# OpenReview forum: "CMC-Bench: Towards a New Paradigm of Visual Signal Compression"
_ICLR.cc/2025/Conference — Submitted to ICLR 2025_

### Official Review · Reviewer_hKWn · 2024-11-01

**Soundness:** 3
**Presentation:** 3
**Contribution:** 3
**Rating:** 8
**Confidence:** 4

**Summary:**

This pioneering work explores the application of LLMs in the context of codec, namely cross-modality compression (CMC). The overall paradigm is innovative, as a combination of I2T+T2I pipeline. Experimental results indicate a satisfying fidelity and extremely low bitrate, which has significant advantages over traditional compression paradigms.

**Strengths:**

(1) Sufficient and comprehensive experiments, namely benchmarks for most current T2I and I2T models. The author has extensively considered mainstream LLMs and explored their optimal combination.

(2) Comparison with traditional codecs such as VVC. Experiments show that CMC has advantages over 7 of the 8 fidelity/quality indicators. This proves the significance of CMC.

(3) Exquisite visualization. Figures 1/2 are both relatively intuitive.

**Weaknesses:**

(1) Ultra-low bitrate needs a better definition. Why 1,000 times? Thousand-fold compression is indeed the highlight of this article, but has there been work in the past that can achieve this bpp? This needs to be further explained, as this is the core novelty of this article.

(2) The author experimented with 12 T2Is but only 6 I2Ts. One or two more I2Ts should be added to the benchmark list.

**Questions:**

Improvement:

(1) I hope the author can add the std of each T2I model. Currently, I only see the mean result on the leaderboard. As you know, the performance of T2I is unstable. Adding std can make the benchmark more complete.

(2) I suggest the author open source all data. Including raw data, test code, and subjective quality labels. As a benchmark, LLM developers need an end-to-end testing pipeline.

---

> ### Author Response · Authors · 2024-11-26
> **Response to Reviewer kYAX [1/1]**
>
> Dear Reviewer kYAX,
>
> Thank you for your positive comments, which greatly motivated us. Regarding the weaknesses and questions you are concerned about, we summarize them as `Q1-Q4`:
>
> >Q1: Defination of 1,000 times compreesion as ULB. Has previous work reached it?
>
> Since the original image is in RGB 8-bit format, i.e. 24 bpp, we set 0.024 bpp as ULB. This 1,000 compression rate, is related to the current development of codecs. Generally speaking, the compression rate of codecs doubles every ten years and is now 500 times, so we set 1,000 as the goal to be achieved.
>
> In the past, some models did achieve 1,000 times [1][2][3], but their results were poor and almost completely different from the original image. It can be said that **CMC is the first to give satisfactory results** at 1,000 times compression.
>
> >Q2: More I2T models for benchmark.
>
> We have added four advanced models, including two I2T (Gemini, InternLM-XComposer) and two T2I (Clandestine, SUPIR). Their performance is shown in the radar map attached in appendix `section A.7.2`. Note that we will maintain this project in the long term, and the performance of mainstream LMMs on CMC-Bench will be updated on the leaderboard every month.
>
> >Q3: std performance for each T2I metric.
>
> We originally wanted to attach the standard deviation in the main text, but had to remove it due to page width. I am happy to update this part here.
>
> | T2I         | Full-FR           | Full-NR           | Image-FR          | Image-NR           |
> |-------------|-------------------|-------------------|-------------------|--------------------|
> | DiffBIR     | 0.268/0.337/0.282 | 0.631/0.695/0.759 | 0.294/0.327/0.278 | 0.491/0.261/0.348  |
> | InstructPix | 0.280/0.374/0.326 | 0.752/0.620/0.736 | 0.268/0.307/0.276 | 0.200/0.272/0.214  |
> | PASD        | 0.283/0.322/0.278 | 0.619/0.501/0.646 | 0.336/0.351/0.284 | 0.580/0.430/0.597  |
> | StableSR    | 0.319/0.315/0.287 | 0.244/0.197/0.282 | 0.323/0.314/0.287 | 0.237/0.194/0.276  |
>
> | T2I   | Full-FR           | Full-NR           | Image-FR          | Image-NR          | Pixel-FR          | Pixel-NR          | Text-FR           | Text-NR            |
> |-------|-------------------|-------------------|-------------------|-------------------|-------------------|-------------------|-------------------|--------------------|
> | Animate   | 0.298/0.274/0.220 | 0.401/0.341/0.513 | 0.283/0.260/0.214 | 0.307/0.294/0.467 | 0.171/0.208/0.175 | 0.558/0.653/0.436 | 0.174/0.180/0.164 | 0.649/0.551/0.606  |
> | Dreamlike | 0.288/0.287/0.259 | 0.395/0.319/0.502 | 0.275/0.274/0.235 | 0.276/0.270/0.411 | 0.163/0.215/0.160 | 0.490/0.601/0.483 | 0.190/0.204/0.186 | 0.625/0.731/0.661  |
> | PG20  | 0.298/0.266/0.241 | 0.506/0.456/0.522 | 0.293/0.239/0.223 | 0.534/0.478/0.580 | 0.173/0.204/0.178 | 0.465/0.489/0.302 | 0.175/0.200/0.196 | 0.625/0.701/0.522  |
> | PG25  | 0.241/0.236/0.233 | 0.632/0.695/0.492 | 0.214/0.216/0.221 | 0.777/0.769/0.567 | 0.173/0.218/0.193 | 0.471/0.510/0.327 | 0.166/0.210/0.199 | 0.500/0.606/0.341  |
> | RealVis  | 0.307/0.313/0.255 | 0.457/0.408/0.543 | 0.305/0.292/0.248 | 0.387/0.351/0.552 | 0.193/0.233/0.188 | 0.448/0.592/0.360 | 0.203/0.221/0.202 | 0.620/0.717/0.494  |
> | SD15  | 0.274/0.271/0.255 | 0.377/0.336/0.428 | 0.273/0.278/0.226 | 0.267/0.225/0.314 | 0.174/0.208/0.151 | 0.534/0.587/0.555 | 0.213/0.220/0.196 | 0.599/0.599/0.616  |
> | SDXL  | 0.297/0.287/0.234 | 0.323/0.309/0.420 | 0.293/0.268/0.226 | 0.266/0.300/0.397 | 0.182/0.221/0.187 | 0.550/0.640/0.392 | 0.179/0.201/0.181 | 0.556/0.575/0.594  |
> | SSD-1B   | 0.289/0.306/0.264 | 0.459/0.405/0.532 | 0.286/0.295/0.253 | 0.403/0.376/0.544 | 0.186/0.215/0.188 | 0.519/0.629/0.368 | 0.192/0.209/0.203 | 0.618/0.677/0.546  |
>
> The three sub-columns in each column of the above table represent the effects of NSI, SCI, and AIGI. Compared with the scores in the main text `Table 4`, we find that the standard deviation of the model is relatively uniform. Basically, there is no encoder/decoder with a huge difference between the upper and lower limits.
>
> >Q4: Open source all data.
>
> Yes, **We will make all the contents of CMC-Bench public, including image data, subjective annotations, and end-to-end test pipelines.** We sincerely hope that it can inspire LLM researchers and drive the evolution of the codec paradigm.
>
> ## Reference
>
> [1] Xue, Naifu, et al. Unifying Generation and Compression: Ultra-low bitrate Image Coding Via Multi-stage Transformer. Arxiv, 2024.
>
> [2] Gao, Junlong, et al. Cross modal compression with variable rate prompt. IEEE TCSVT, 2024.
>
> [3] Gao, Junlong, et al. Rate-distortion optimized cross modal compression with multiple domains. IEEE TCSVT, 2024.
>
> Best Regards
>
> CMC-Bench Author Team

---

### Official Review · Reviewer_1Puj · 2024-11-01

**Soundness:** 3
**Presentation:** 3
**Contribution:** 3
**Rating:** 6
**Confidence:** 4

**Summary:**

The paper introduces CMC-Bench, a benchmark for Cross Modality Compression (CMC) in image compression. Utilizing advancements in Large Multimodal Models (LMMs), the authors explore an Image-to-Text-to-Image (I2T-T2I) approach, where visual signals are first transformed into text and then regenerated into images, achieving compression rates as high as 10,000 times compared to traditional codecs. However, the CMC paradigm faces challenges in consistency and perception fidelity at ultra-low bitrates. CMC-Bench evaluates 6 I2T and 12 T2I models across 58,000 images, covering consistency and perception dimensions with 160,000 subjective human annotations, providing a comprehensive dataset for future codec development. Experimental results indicate that CMC has the potential to outperform traditional codecs but requires further refinement in handling Screen Content Images (SCI) and balancing consistency and perception.

**Strengths:**

This paper introduces a new benchmark platform for Cross Modality Compression (CMC) called CMC-Bench, offering a fresh approach and guiding direction for future ultra-low bitrate image compression. The paper includes 58,000 images and 160,000 human annotations, offering a large-scale and comprehensive evaluation. Furthermore, authors presents rigorous comparative experiments with existing traditional codecs, demonstrating the advantages of CMC, which is highly persuasive. Although it does not propose a complete solution, the dataset and evaluation metrics provided in this work are likely to inspire further research.

**Weaknesses:**

Some details require further discussion, such as the balance between consistency and perceptual quality. Additionally, certain expressions and the structure of the paper would benefit from further refinement and polishing.

**Questions:**

1.	For each column in Table 1, it is recommended to provide the full name and corresponding unit (e.g., "dis") in the main text or caption.

2.	According to the description in the text, the radar chart on the right side of Figure 5 represents "{GPT-4o + 12 different T2I models}," but it appears there are only six methods shown on the chart. The description is somewhat confusing.

3.	In Section 4.2, the first sentence of the first paragraph should be changed to “Figure 5 shows the performance of 6 I2T models as encoders and 12 T2I models as decoders in image compression.”

4.	The three colors in the histogram in Figure A3 are not all explained within the figure.

5.	I am curious about how the authors balanced consistency and perceptual quality when establishing the evaluation metrics, or if any experiments or considerations were conducted regarding this. Also, did the authors perform any ablation studies to determine the final weights of each evaluation metric?

6.	It seems the authors could further adjust the organization of the main text and appendix to prioritize more important information in the main body.

7.	The authors have indeed undertaken extensive integration and experiments related to previous data and evaluation methods in establishing the CMC-Bench standardized evaluation platform for I2T-T2I. However, this work seems more like an integration of previous research efforts, including the sources of the new dataset and the evaluation metrics, which resemble normalized methods from prior I2T-T2I tasks. Are the authors concerned that this paper’s originality might be questioned?

---

> ### Author Response · Authors · 2024-11-26
> **Response to Reviewer 1Puj [1/2]**
>
> Dear Reviewer 1Puj,
>
> Thank you for the positive comments on CMC-Bnech. In particular, thanks for your **character-level review** of our article. We are honored to receive **such a responsible review**. We will reply to your Question from `Q1-Q7`, and then to Weakness as `Q8`.
>
> >Q1: Full name and corresponding unit in Table 1.
>
> Sure. We have **added** the full name of `dis/ref` in the title of `Table 1`. Please check the latest manuscript.
>
> >Q2: Figure 5 represents 12 models but only six methods are shown on the chart.
>
> For aesthetic reasons, it would be messy to plot all 12 T2I models on the radar chart. Thus we only visualized 6 representative models in `Figure 5`. The detailed performance of all models is placed in `Table 3` below.
>
> According to your suggestion, we also **changed** the title of `Figure 5` to 6 models.
>
> >Q3: Typo in Section 4.2.
>
> Thanks for addressing such a hidden typo! We have corrected it in the latest manuscript.
>
> >Q4: The three colors in the histogram in Figure A3 are not all explained within the figure.
>
> In fact, there are only two colors. For better visualization, we set the transparency of them to 0.5. The third color you see is the overlap of the two colors.
>
> >Q5: Balancing consistency and perceptual quality when establishing the evaluation metrics, and ablation to determine the final weights
>
> We set the ratio of consistency and perceptual quality to 2:1 for the following reasons: (1) Consistency is more important than perception. As it is feasible to enhance low-quality, high-consistency images (like using super-resolution); while improving a high-quality, low-consistency image is challenging. (2) Consistency and perception have different fluctuation ranges. Here we use TOPIQ-FR and TOPIQ-NR respectively. `Table 3/4` shows that the former has a smaller fluctuation range. Therefore, for fairness, its weight needs to be increased.
>
> We **added ablation experiments** to demonstrate why equal (1:1) or higher (3:1) weights are incorrect. First, `Figure A8-A13` shows that under ULB (the first two rows of images), the results obtained by methods such as `RealVis/SSD-1B` are consistent with the original image, while the methods focusing on Perception, such as PG20/PG25, perform poorly. Under ELB (the last two rows), the results obtained by all methods are not completely consistent with the original image, so `PG25/PG20` should be ranked higher by humans.
>
> | Rank | ULB  | ULB | ULB | ELB | ELB | ELB |
> |-----------|-----------|-----------------|-----------|-----------|-----------------|------------|
> |Ratio| 1:1         | **2:1**               | 3:1         | 1:1         | **2:1**               | 3:1          |
> |1| RealVis   | RealVis         | RealVis   | PG25      | PG25            | RealVis    |
> |2| PG25      | SSD-1B          | SSD-1B    | RealVis   | RealVis         | PG25       |
> |3| PG20      | PG20            | SDXL      | PG20      | PG20            | PG20       |
> |4| SSD-1B    | SDXL            | PG20      | SSD-1B    | SSD-1B          | SSD-1B     |
> |5| SDXL      | Dreamlike       | Dreamlike | SDXL      | SDXL            | SDXL       |
> |6| Animate   | PG25            | SD15      | Animate   | Dreamlike       | Dreamlike  |
> |7| Dreamlike | Animate         | Animate   | Dreamlike | Animate         | Animate    |
> |8| SD15      | SD15            | PG25      | SD15      | SD15            | SD15       |
>
> The leaderboard above shows that if a 1:1 weight is used, `PG25/PG20` will rank second/third respectively under ULB, which places **too much emphasis on Perception**; if a 3:1 weight is used, `RealVis` will win the championship under ELB, which places **too much emphasis on Consistency.** Therefore, we believe this ratio is most consistent with the subjective evaluation results of humans.
>
> >Q6: Further adjust the organization of the main text and appendix.
>
> Yes, we have made revisions. We have **added more experimental settings** in the main text and moved some **data analysis** to the appendix `section A.7`.

---

> ### Author Response · Authors · 2024-11-26
> **Response to Reviewer 1Puj [2/2]**
>
> >Q7: Are the authors concerned that this paper’s originality might be questioned? (looks like a combination of existing works)
>
> In general, CMC-Bench has certain novelty for both issues you are concerned about. (1) Data source. CMC-Bench has **only 1,000 reference images** taken from the previous dataset, and **58,000 distorted images** are all generated by itself. Most importantly, these images have **160,000 quality annotations**, all of which come from human subjective evaluation. Therefore, **most of the dataset is original.** (2) Whether it is a combination of existing models. CMC-Bench is not a simple connection of existing models. We have comprehensive considerations on bitrate, parameter settings, and reference image formats, such as the verification of parameter λ in `Figure A5`. If I2T+T2I are directly connected without any operation, you can refer to the results of PICs [1] and Image2Paragraph [2], which are disastrous. In short, CMC-bench provides **an effective combination** and surpasses VVC for the first time, which is original.
>
> Most importantly, we propose an **end-to-end evaluation pipeline, not just a compression algorithm.** Any LMM developer (both T2I and I2T) can plug in and test performance easily. It can expand the influence of the codec community and directly benefit from the rapid development of LLM. I think this contribution is more important than one codec.
>
> >Q8: Further discussion on the balance between consistency and perceptual quality, and polish certain expressions.
>
> Thank you for your suggestion. We have incorporated the response in `Q5`, namely **consistency-perception trade-off**, [3][4] into the manuscript, you may check appendix `section A.7.3`. Regarding writing, we invited a native English-speaking professor from Canada to revise the paper at the character level.
>
> ## Reference
>
> [1] Lei, Eric, et al. Text+ sketch: Image compression at ultra-low rates. arXiv preprint arXiv:2307.01944. 2023.
>
> [2] ShowLab. Image2Paragraph. https://github.com/showlab/Image2Paragraph. 2023.
>
> [3] Blau, Yochai, et al. The perception-distortion tradeoff.In Proceedings of the IEEE conference on computer vision and pattern recognition, pages 6228–6237, 2018.
>
> [4] Blau, Yochai, et al. Rethinking lossy compression: The rate-distortion-perception tradeoff.In International Conference on Machine Learning, pages 675–685. PMLR, 2019.
>
> Best Regards
>
> CMC-Bench Author Team

---

### Official Review · Reviewer_CJXY · 2024-11-03

**Soundness:** 3
**Presentation:** 3
**Contribution:** 3
**Rating:** 5
**Confidence:** 4

**Summary:**

This paper builds a benchmark dataset for Cross Modality Compression (CMC), which aims at achieving much better compression at extremely low bitrates. This benchmark covers tens thousands of image samples and evaluates on 6 Image2Text models and 12 Text2Image models for CMC task. Human annotations are given to evaluate the subjective preference of compression results. There are four modes tested for cross modality compression in this paper, including text-conditioned compression (Text-mode) pixel-conditioned compression (Pixel-mode) traditional codec-based generative compression (Image-mode), and a combination of them all (Full-mode). After analyzing the collected dataset, this paper concludes that without dedicated training for compression, combinations of several advanced I2T and T2I models have already surpassed traditional codecs in multiple aspects.

**Strengths:**

I appreciate the effort that the authors put in annotating subjective scores and building such a relatively large dataset for image compression.

**Weaknesses:**

1. Throughout the paper, I did not find any non-trivial insight from the proposed benchmark for the whole community. The main conclusion of the proposed benchmark assessment is that: in the task of ultra-low rate compression, codecs based on generative models (e.g., T2I models) could perform better than traditional codecs in many aspects. However, actually it is almost a common sense in the compression community, I would suggest the authors to think deeper so that it can get some valuable conclusions, such as “which type of T2I architecture would perform better for CMC task, auto-regressive or denoising diffusion?”

2. Some sentences in this paper are overclaimed. For example, in abstract, it is written as
“CMC has certain defects in consistency with the original image consistency with the original image and perceptual quality. To address this problem, we introduced CMC-bench…”
How could the benchmark address the issue of inconsistency in CMC. There is no technical algorithm proposed in this paper to address this issue, neither any direct insight in the proposed benchmark dataset that guides the community to address the inconsistency issue.

3. I have some doubts on the reliability of the proposed benchmark dataset. For example, in figure7, the performance of GPT-4o +RealVis fluctuates in FID-bpp. Sometimes FID increases and sometimes FID decreases as bitrates increases.

**Questions:**

As mentioned above, I am wondering is there any specific example that proves the proposed benchmark dataset can help resolve the inconsistency issue of cross-modality compression?

Except the issues I mentioned in the Weakness section, I also wonder whether there is any bias during the human annotation. Some experimental details should be added to ensure the subjective scores are reliable and unbiased.

**Details Of Ethics Concerns:**

I will have no ethics concerns if all the interviewers are fairly paid or rewarded when they give subjective annotations.

---

> ### Author Response · Authors · 2024-11-26
> **Response to Reviewer CJXY [1/2]**
>
> Dear Reviewer CJXY,
>
> We appreciate your constructive feedback. Your guidance is crucial in advancing our work, especially in **giving some in-depth insight beyond the experiment.** On the technical level, we have summarized your opinion about weakness as `Q1-Q3` and questions as `Q4-Q5`, then responded to them one by one:
>
> >Q1: Give some valuable non-trivial insight, such as “which type of T2I architecture would perform better for CMC task.”
>
> Thank you for your feedback. Based on your request, we have drawn the following conclusions beyond that "CMC is superior to traditional codecs":
>
> 1. Is the AR or the Diffusion architecture better for CMC tasks?
> Generally speaking, the Diffusion architecture is better. **This is not because the AR performance metrics are insufficient, but because generating pixels one by one leads to a higher complexity.** We acknowledge that the AR model is more controllable and may sometimes achieve better performance than the Diffusion model, but it requires a massive amount of parameters for the T2I model and longer decoding times. Work [1] shows that AR generation of a 384*384 image takes more than 6 seconds, which is enough time to generate several 1024-resolution images with a Diffusion model whose size is less than 1B. Considering the real-time nature of visual communication, we believe that the Diffusion model is more suitable for CMC.
> 2. Among the Diffusion architectures, which one is better?
> This requires a case-by-case discussion, including the Restoration, SDXL, and SD architectures. The leaderboard in `Figure 6` shows that on the ULB, the order is `Restoration > SD > SDXL`, because Restoration fine-tunes based on the original image, SD uses the original image as the starting point to draw, and SDXL, compared to SD, has more freedom to make changes, which can lead to excessive modifications. On the ELB, the order is `SDXL > SD`, because without a reference image, Restoration is directly unavailable. Compared to SDXL's aggressive strategy, which becomes an advantage, it can produce more details.
>
> Due to space limitations, we decided to **add these analyses** to the appendix `section A.7.1` to inspire the codec community's thinking on this issue.
>
> >Q2: Technical algorithm, or direct insight in the proposed benchmark to address the inconsistency issue.
>
> Yes, as a benchmark paper, its focus is indeed not on the technical algorithms, but it **still gives direct insights for inconsistency problems.** Firstly, CMC-Bench has proven that the cascaded approach of I2T + T2I can directly surpass the state-of-the-art VVC. Note that this is different from the previous conclusion that generative codecs are superior to traditional codecs. This is because previous methods required fine-tuning processes such as VQ-VAE-2 [2] and DDIM [3], whereas the approach based on LLM is **completely training-free.** Secondly, CMC-Bench has obtained effective evaluation standards through a large amount of human subjective preference data, as seen in `Table 2`. Take consistency as an example, the lpips commonly used to evaluate codec performance have a correlation of less than 0.6 with human subjective scores, while the data-driven by TOPIQ has a correlation with human preferences that exceeds 0.9. Compared to traditional performance indicators, it **provides a more accurate direction for optimization** in the inconsistency problem.
>
> To avoid the suspicion of overclaiming, we have **modified the expression** of the sentence in the abstract.
>
> >Q3: Benchmark reliability. For example, the increasing/decreasing in Figure 7.
>
> This is an interesting question. The fluctuation of the FID is due to the four different modes of CMC, of which the first two modes (Text and Pixel, i.e., the ELB in Q1) grant the T2I model a greater degree of editing freedom, at 0.8; while the latter two modes (Image and Full, i.e., the ULB in Q1), in order to improve consistency, reduce the freedom to 0.5. As we emphasize in our paper, **consistency and perception are a dilemma, and the codec task must first ensure consistency**, so we have made this setting of permissions. Therefore, you will see that most of the consistency metrics in `Figure 7` increase with the bpp, while perception shows an unusual drop between the second and third points.
>
> >Q4: Specific example about solving the inconsistency issue.
>
> Yes, we have **added examples** in `Figure A17` of the Appendix, please check the main PDF. The figure shows that while the consistency of CMC may not be perfect, it still significantly outperforms VVC. **Note that CMC is targeted at Ultra-low bitrate, although it still has shortcomings, other models are almost unable to work in such a challenging scenario.** Therefore, we believe that CMC provides the first acceptable baseline solution for inconsistency issues.

---

> ### Author Response · Authors · 2024-11-26
> **Response to Reviewer CJXY [2/2]**
>
> >Q5: Bias during the human annotation.
>
> Yes, bias is an important factor. Therefore, we calculated the correlation of subjective scores between the 20 subjects, represented by the Spearman Rank Correlation Coefficient (SRCC). Due to space limitations, we only present the results for the first 10 subjects as shown in the table below:
>
> | | Subject 1 | Subject 2 | Subject 3 | Subject 4 | Subject 5 | Subject 6 | Subject 7 | Subject 8 | Subject 9 | Subject 10 |
> |---|---|---|---|---|---|---|---|---|---|---|
> | **Subject  1** |  | 0.7655 | 0.7813 | 0.7782 | 0.7164 | 0.7657 | 0.7717 | 0.7311 | 0.8181 | 0.7403 |
> | **Subject  2** | 0.7655 |  | 0.8017 | 0.7955 | 0.7157 | 0.7967 | 0.7943 | 0.7756 | 0.8571 | 0.8277 |
> | **Subject  3** | 0.7813 | 0.8017 |  | 0.8152 | 0.7567 | 0.7700 | 0.8063 | 0.7997 | 0.8475 | 0.7731 |
> | **Subject  4** | 0.7782 | 0.7955 | 0.8152 |  | 0.7523 | 0.7729 | 0.8225 | 0.8015 | 0.8508 | 0.7838 |
> | **Subject  5** | 0.7164 | 0.7157 | 0.7567 | 0.7523 |  | 0.6992 | 0.7206 | 0.7139 | 0.7831 | 0.6868 |
> | **Subject  6** | 0.7657 | 0.7967 | 0.7700 | 0.7729 | 0.6992 |  | 0.7302 | 0.7803 | 0.8339 | 0.7915 |
> | **Subject  7** | 0.7717 | 0.7943 | 0.8063 | 0.8225 | 0.7206 | 0.7302 |  | 0.7966 | 0.8385 | 0.7836 |
> | **Subject  8** | 0.7311 | 0.7756 | 0.7997 | 0.8015 | 0.7139 | 0.7803 | 0.7966 |  | 0.8215 | 0.7849 |
> | **Subject  9** | 0.8181 | 0.8571 | 0.8475 | 0.8508 | 0.7831 | 0.8339 | 0.8385 | 0.8215 |  | 0.8366 |
> | **Subject  10** | 0.7403 | 0.8277 | 0.7731 | 0.7838 | 0.6868 | 0.7915 | 0.7836 | 0.7849 | 0.8366 |  |
>
>
> According to the ITU standard [4], an SRCC of 0.6 or higher indicates that a subject has been diligent in their scoring, while an SRCC of 0.7 or higher indicates that two subjects have similar opinions. An SRCC of 0.9 or higher is considered too high (lacking differences). The table above shows that the SRCC for the subjects is **within an acceptable range, with no outliers that are too low and no duplicated data that is too high.** Therefore, the bias issue has been effectively addressed.
>
> ## Reference
>
> [1] Sun, Peize, et al. Autoregressive Model Beats Diffusion: Llama for Scalable Image Generation. arXiv. 2024.
>
> [2] Mentzer, Fabian, et al. "High-fidelity generative image compression." Advances in Neural Information Processing Systems 33. 2020.
>
> [3] Yang, Ruihan, and Stephan Mandt. Lossy image compression with conditional diffusion models. Advances in Neural Information Processing Systems 36. 2024.
>
> [4] I.T.Union. Methodology for the subjective assessment of the quality of television pictures. ITU-R Recommendation BT.500-11, 2002.
>
> Best Regards
>
> CMC-Bench Author Team

---

### Official Review · Reviewer_kYAX · 2024-11-06

**Soundness:** 3
**Presentation:** 4
**Contribution:** 2
**Rating:** 5
**Confidence:** 5

**Summary:**

This paper introduces CMC-Bench, a benchmark designed to evaluate ultra-low bitrate image compression using a Cross Modality Compression (CMC) approach with Image-to-Text (I2T) and Text-to-Image (T2I) models. To assess the coding performance of CMC-based codecs, four collaboration modes are presented along with two indicators: consistency and perception. Experimental results show that some combinations of existing I2T and T2I models outperform traditional codecs in terms of some quality metrics at ultra-low bitrates.

**Strengths:**

1. The idea of conducting a benchmark to evaluate CMC-based codecs is valuable.
2. This paper provides a comprehensive set of experiments studying various combinations of I2T and T2I models.

**Weaknesses:**

1. This paper does not propose a new model but instead (1) creates a mixed dataset by collecting images from several existing datasets and (2) combines existing I2T and T2I models to evaluate their coding performance. The technical novelty of this paper is therefore questionable.
2. The reported -FR and -NR values are the weighted average of multiple metrics, but it is unclear how the proposed weighting ensures a reasonable assessment.
3. In Fig. 6, the gap between the upper and lower bounds of most variation bars exceeds 1, with some even approaching 2. Given that the MOS scale ranges only from 1 to 5, is such a large gap reasonable? Can reliable conclusions be drawn from this?
4. Although the authors claim that some combinations of existing I2T and T2I models outperform traditional codecs at ultra-low bitrates, this is only demonstrated for specific objective functions. A MOS comparison and subjective evaluation between traditional codecs and CMC-based codecs are not provided.
5. PSNR result is not reported.
6. The RD curve of GPT-4o + RealVis in Fig.7 looks very weird.
7. It appears that all the data used for training and evaluating the FR/NR quality indicators are generated using GPT-4 (OpenAI, 2023) and 12 different T2I models. However, this does not include images generated by other methods, such as traditional codecs or any learning-based codecs. Wouldn't the indicators trained on this data be biased?

**Questions:**

1. The method for obtaining the compression rate (CR) values in Section 3.2 is unclear. Different codecs and rate points are expected to yield significantly varied results.

2. The reported -FR and -NR values are the weighted average of multiple metrics, but it is unclear how the proposed weighting ensures a reasonable assessment.

3. In Fig. 6, the gap between the upper and lower bounds of most variation bars exceeds 1, with some even approaching 2. Given that the MOS scale ranges only from 1 to 5, is such a large gap reasonable? Can reliable conclusions be drawn from this? It is also unclear how the rankings in Fig. 6 were determined given the substantial variation in MOS scores.

4. It is unclear how this paper deal with the randomness in CMC-based codecs.

5. According to the visualization results shown in Figs. 8-13, although the decoded images appear perceptually good, they differ significantly from the input images. Moreover, the RD curves suggest that CMC-based codecs are effective only at extremely low rates and cannot be extended to higher rates. Even if CMC-based codecs outperform traditional codecs in some objective evaluations, do they truly have practical applications in real-world compression scenarios? Even if CMC-based codecs outperform traditional codecs in some objective evaluations at extremely low rates, do they really have practical applications in real-world compression scenarios?

6. It appears that all the data used for training and evaluating the FR/NR quality indicators are generated using GPT-4 (OpenAI, 2023) and 12 different T2I models. However, this does not include images generated by other methods, such as traditional codecs or any learning-based codecs. Wouldn't the indicators trained on this data be biased?

---

> ### Author Response · Authors · 2024-11-26
> **Response to Reviewer kYAX [1/2]**
>
> Dear Reviewer kYAX,
>
> Thank you for your detailed comments, and your recognition of the benchmark section. Since the Weakness and Question you mentioned have some overlap, we have merged them into 10 questions and responded to them point by point.
>
> >Q1: This paper does not propose a new model but instead (1) creates a mixed dataset by collecting images from several existing datasets and (2) combines existing I2T and T2I models to evaluate their coding performance.
>
> Thanks for your question. Actually, most of the CMC dataset is original. (1) **Only 1,000 reference** images were taken from the existing dataset you mentioned, while **58,000 distorted images were all generated by ourselves.** Most importantly, these images have **160,000 quality annotations**, all of which come from human subjective evaluation we conducted.  (2) CMC-Bench is not a simple concatenation of existing models. We have comprehensive considerations in terms of bitrate, parameter settings, and the format of reference images, such as the validation of the parameter λ in `Figure A5`. If the I2T+T2I connection is directly made without any operations, you can refer to the negative results of PICs [1] and Image2Paragraph [2]. In summary, CMC-Bench has provided an **effective combination instead of a simple connection** and has surpassed VVC with novelty.
>
> >Q2: The reported -FR and -NR values are the weighted average of multiple metrics, but it is unclear how the proposed weighting ensures a reasonable assessment.
>
> Our weighting comes from two indicators, TOPIQ-FR and NR, in a 2:1 ratio. Their combination has been strictly verified, see Q1. The rationality is reflected in (1) both indicators are consistent with human subjective preferences, see `Table 2`, and the correlation coefficient is above 0.9; (2) the weight ratio is reliable, see `Table 3/4`, where FR and NR have different fluctuation ranges, the former has a smaller fluctuation range. Thus we apply this ratio for a fair comparison.
>
> >Q3: Is the large gap between the upper and lower bounds reasonable? Can reliable conclusions be drawn from this? How the rankings in Fig. 6 were determined?
>
> We apologize for any misunderstanding we may have caused. In fact, the discrepancy stems from changes in the λ parameter, not instability within the model itself. The bar chart and table differ in that the bar chart tested multiple configurations of λ, ranging from 0.2 to 0.8, in order to find the optimal setting; the table, however, is based on a single λ value, as the optimal obtained above, to ensure fairness. As  shown in `Figure A6`, the variation in parameters from 0.2 to 0.8 is the reason for the discrepancy; once fixed, the model's performance is much more stable, whose std is shown in the following table:
>
> | std       | Full-FR | Full-NR | Image-FR | Image-NR | Pixel-FR | Pixel-NR | Text-FR | Text-NR |
> |-----------|---------|---------|----------|----------|----------|----------|---------|---------|
> | Animate   | 0.2596  | 0.4278  | 0.2485   | 0.3671   | 0.1837   | 0.5377   | 0.1718  | 0.6024  |
> | Dreamlike | 0.2761  | 0.4150  | 0.2587   | 0.3282   | 0.1774   | 0.5205   | 0.1926  | 0.6712  |
> | PG20      | 0.2656  | 0.4974  | 0.2488   | 0.5356   | 0.1843   | 0.4070   | 0.1909  | 0.6066  |
> | PG25      | 0.2363  | 0.5949  | 0.2174   | 0.6906   | 0.1945   | 0.4251   | 0.1924  | 0.4682  |
> | RealVis   | 0.2880  | 0.4767  | 0.2783   | 0.4422   | 0.2030   | 0.4560   | 0.2080  | 0.5987  |
> | SD15      | 0.2655  | 0.3851  | 0.2557   | 0.2732   | 0.1750   | 0.5583   | 0.2083  | 0.6058  |
> | SDXL      | 0.2688  | 0.3576  | 0.2587   | 0.3286   | 0.1957   | 0.5138   | 0.1864  | 0.5769  |
> | SSD-1B    | 0.2841  | 0.4720  | 0.2755   | 0.4513   | 0.1955   | 0.4916   | 0.2015  | 0.6069  |
>
> It can be seen that under the same λ, **std is not as large as the figure error bar. As we fixed the λ during evaluation, the reliability is ensured.** The rankings in `Figure 6` are derived from a 2:1 mix of Consistency and Perception, as described in `Q2`.
>
> >Q4: Subjective evaluation between traditional and CMC-based codecs.
>
> The reason why we use the objective function is for end-to-end evaluation, to ensure that the community can reproduce our benchmark. Therefore, we first let people score, and then use the objective function to fit. (Table 2 shows that the correlation between the two is 0.94, which is a good fit, that is, objective evaluation is able to replace subjective)
>
> To further prove it, we also **added subjective evaluation** as you requested, comparing advanced codecs and CMC, see appendix `Figure A9` and `section A.7.4`. We show the results of the two codecs and let five experts choose their preferences. The majority's choice is considered better. Overall, CMC still has a greater advantage.

---

> ### Author Response · Authors · 2024-11-26
> **Response to Reviewer kYAX [2/2]**
>
> >Q5: PSNR result is not reported.
>
> The experiment follows the evaluation principle of covering all high-mid-low levels. Since SSIM is already available for low-level, PSNR is not shown in the main text.
>
> We have **added the PSNR results** as you requested, see `Figure A10` in the appendix.
>
> >Q6: The RD curve of GPT-4o + RealVis in Fig.7 looks weird.
>
> This is an interesting question. The curve in `Figure A4` appears unusual because **the y-axis range is small**. When compared to the -axis in `Figure A5` below, it becomes clear that the changes are very weak. **We are demonstrating that as the number of words in the I2T model increases, the performance changes very little, almost like minor oscillations.** This ensures that the length of the text will not affect the fairness of the benchmark.
>
> >Q7: The training does not include images generated by other methods, such as traditional codecs or any learning-based codecs. Wouldn't the indicators trained on this data be biased?
>
> Yes, we used `GPT-4o+12 T2I` as training data for the following reasons: (1) No other I2Ts: The differences between I2Ts are not large, as shown in the annotations on `Figure 5`. The performance of different I2Ts differs by at most 0.05, and the performance of different T2Is differs by 1. Therefore, we did not use data like `Qwen+12 T2I` or `LLaVA+12 T2I`. (2) No other codec: The initial weights of the IQA model are already trained on compressed images (JPEG, JP2K, VVC), thus there is prior knowledge. From Table 2, both FR and NR exceed 0.9. With such high performance, there is almost no deviation in the IQA field. [3]
>
> >Q8: The method for obtaining the compression rate.
>
> We set the compression rate based on VVC `QP=53`. You can refer to the description in Appendix `Section A.6`.
>
> >Q9: Randomness in CMC-based codecs.
>
> As we replied in `Q3`, the randomness of CMC is not that serious. The larger error bar is because we introduced different λ parameters in the training data. We sincerely apologize again for the misunderstanding.
>
> >Q10: If CMC-based codecs outperform traditional codecs in objective evaluations? Even if CMC outperforms others, do they really have practical applications in real-world compression scenarios?
>
> **Outperform traditional metric:** Yes. Note this work faces an extremely compressed scenario. (In other words, it is unfair to compare the fidelity of two images when CMC is compressed 10,000 times and the traditional method is compressed 100 times. We need to compare them at the same bpps). CMC has the following two fidelity advantages over existing work: (1) At a compression ratio of 1,000 times, namely 0.024bpp, the traditional method can only give the outline of the image, while CMC can restore the approximate information. In other words, even if CMC has a small probability of "you are not yourself", the traditional compression will most likely produce "you are pixel"; (2) Under a compression ratio of 10,000 times, namely 0.002bpp, no traditional method can achieve this extreme condition, while CMC can at least semantically reconstruct the image and give the main object, background, color and other information consistent with the original image.
>
> **Real-world application:** Currently, CMC can be used in the following two scenarios. (1) Communication: In severe scenarios, channel resources are extremely limited, such as deep sea and space; or there are too many devices, that is, hundreds of devices in IoT share a local area network. In this case, traditional compression methods cannot adapt to such a low bit rate, and image communication can only be carried out through CMC; (2) Storage: According to statistics from mainstream social media, 10% of visual information contributes to 99% of views, and most images are "junk data". For these images that are rarely clicked but not suitable for deletion, their storage will consume considerable resources. Therefore, they can be compressed in CMC format and decompressed when needed. With the advancement of LMM, models with lower complexity have emerged in recent years. We believe that CMC can move from these two applications with low latency requirements to future real-time scenarios.
>
> ## Reference
>
> [1] Lei, Eric, et al. Text+ sketch: Image compression at ultra-low rates. arXiv preprint arXiv:2307.01944. 2023.
>
> [2] ShowLab. Image2Paragraph. https://github.com/showlab/Image2Paragraph. 2023.
>
> [3] I.T.Union. Methodology for the subjective assessment of the quality of television pictures. ITU-R Recommendation BT.500-11, 2002.
>
> Best Regards
>
> CMC-Bench Author Team

---

### Meta-Review · Area_Chair_EeEa · 2024-12-17

**Metareview:**

This paper attempts to evaluate several potential approaches to using large multimodal models (LMMs) for image compression. It is a compelling idea to represent images simply using a textual description – it could potentially yield ultra-low bit rate image compression methods.

The authors collect an impressive amount of data using various multimodal models, such as image-to-text (I2T) and text-to-image (T2I) models, and evaluate the size of the textual representation (and/or a severely downsampled version of the image) against the reconstructed image using subjective (human) experiments as well as an adapted version of TOPIQ, an image quality assessment method.

After reading the paper and digesting the comments of the reviewers as well as the authors' rebuttal, I have decided to reject this paper, for the following reasons:

1. The authors focus on ranking various models used for encoding and decoding against each other. In my opinion, this is "putting the cart before the horse".

   Many in the image compression community have justifiable doubts around the robustness of the proposed methods, as the authors admit in the introduction. I.e.: it is clear that there are individual images that might be extremely well represented using a short text, but would this method reliably give compression gains for arbitrary images? The results shown in the paper do seem to point towards a "no" to this question (for example: the picture of the woman in Fig. 1 representing a different person, relatively large error bars in Fig. 6, erratic rate–quality plots in Figs. 7 and A4), but the authors neglect discussing this.

   I tend to agree with Reviewer CJXY: The paper's focus should be on identifying what robustness issues there are, what properties of I2T and T2I models give rise to them, and generate insights that last beyond any individual LMM. This would enable the community to learn something, and make progress, independent of the fast-paced development of LMMs. Once we are in a regime of more incremental research, it makes sense to focus on a more quantitative evaluation as the authors do here.

2. I further share methodical concerns with Reviewer kYAX: For example, I am not convinced that fine-tuning TOPIQ yields an unbiased evaluation method. When we consider conceptually novel compression methods such as here, subjective evaluation is more reliable, since objective IQA methods are often only validated with respect to specific classes of compression/generation methods and may not generalize well enough. It would be more appropriate to focus on subjective evaluation, or spend more time validating and justifying specific metrics for the use in this context.

3. The language of the paper is muddled and imprecise, which makes details difficult to understand. For example, the abstract talks about using I2T and T2I models, but in fact, authors also use downsampled image representations for compression. As another example, I find the section on page 6 starting in line 298 almost impossible to parse – after several attempts, it is still unclear to me what the authors exactly did to come up with an evaluation method. There are more examples.

With this said, I don't believe all is lost – for example, for a future paper, the authors could use the data they gathered with human experiments, and mine it to generate more qualitative insights.

**Additional Comments On Reviewer Discussion:**

There was a reasonable amount of interaction between reviewers and authors. I discounted the positive rating from Reviewer hKWn. I think they are missing several issues with this work.

---

### Decision · Program_Chairs · 2025-01-22

Reject